# Measuring Invariance in Representation Learning: A Robust Evaluation Framework

## Abstract

Distribution shifts pose a significant challenge to the reliable deployment of machine learning models. Invariant representation learning aims to mitigate this challenge by learning feature spaces that remain invariant across diverse out-of-distribution (OOD) scenarios. However, a critical gap exists in directly and efficiently evaluating the true invariance of learned representations across varied environments. To address this, we introduce DRIC, a novel and computationally efficient criterion designed for the direct assessment of invariant representation performance. DRIC establishes a formal link between the conditional expectation of invariant predictors and environmental diversity through the density ratio, providing a theoretically sound and practical evaluation framework. We validate the effectiveness and robustness of DRIC through extensive numerical experiments on both synthetic and real-world datasets, demonstrating its utility in quantifying and comparing the invariance of learned representations, ultimately contributing to the development of more robust machine learning models.

## 1 Introduction

The assumption of independently and identically distributed (i.i.d.) data has been a standard assumption in statistical machine learning. However, in real-world scenarios, data can originate from diverse environments, potentially violating the assumption of homogeneous distribution (Ahuja et al., 2021; Cai et al., 2023). The Empirical Risk Minimization (ERM) (Yang et al., 2023) has proven effective in solving this problem by considering the average loss across all training environments. However, if the training environments themselves exhibit heterogeneity, the resulting model may struggle to generalize to unseen environments. This sensitivity to training environments introduces model instability. Thus, many invariant learning methods are proposed to enhance generalization across unseen environments.

The significance of invariant learning (IL) lies in its ability to enhance the robustness of machine learning models. By incorporating assumptions of invariance properties (Peters et al., 2016) in learning invariant representations, these models can better handle scenarios where data may vary in different environments. Examples of invariant learning include *Domain-Adversarial Neural Network (DANN)* developed by Ganin et al. (2015) and Ganin et al. (2016) for domain adaptation. Li et al. (2018) proposed an end-to-end conditional invariant deep domain generalization approach by leveraging deep neural networks for domain-invariant representation learning. Motiian et al. (2017) introduced a deep model augmenting the classification and contrastive semantic alignment loss to address the domain generalization problem. Mitrovic et al. (2020) proposed *Representation Learning via Invariant Causal Mechanisms (RELIC)* for the classification task by enforcing the preservation of the underlying probability across different domains. *Domain-Specific Adversarial Network (DSAN)* from Stojanov et al. (2021) offered broader applicability by assuming the invariance of the conditional distribution of the outcomes. Yao et al. (2022) proposed *LISA*, a simple mixup-based technique to learn invariant predictors via selective augmentation. Arjovsky et al. (2019) introduced *Invariant Risk Minimization (IRM)* by simply assuming the invariance of the conditional expectation of the outcome given the invariant representation, which is applicable to diverse learning tasks.

There are extensive literature on IL methods, see Section 5. A representation is a mapping $\phi : \mathcal{X} \to \mathcal{Z}$ from covariates to a latent space; IL seeks a *domain-invariant* $\phi$ for which the predictive relation between $Z = \phi(X)$ and $Y$ is identical across environments. In practice, enforcing invariance often lowers training accuracy yet can improve OOD test accuracy up to a point. If over-enforced, they degrade both, so model selection must balance accuracy and invariance. The lack of standardized assessment makes it challenging to evaluate the overall performance of such methods and to strike this balance. To address this issue, our study develops a robust assessment that measures representation-level invariance and thereby guides the selection of an optimal invariant representation.

In this paper, we propose a quantity called the *Density Ratio based Representation Invariance Criterion* (DRIC) to serve as a robust metric of invariance for all IL methods that employ the invariance property 2.2. The overall DRIC workflow is shown in Figure 1, comprising candidate invariant models, a DRIC evaluator, and optimal representation selection. DRIC is the first environment-agnostic, normalized metric that is comparable across IL methods and datasets with the following contributions.

- We introduce a density-ratio–based, environment-agnostic, normalized metric that directly measures representation-level invariance and is comparable across IL methods and datasets.
- We give a simple, classifier-based plug-in estimator and a representation selection workflow.
- We provide theoretical guarantees for DRIC and its estimator's convergence, along with information lower bounds balancing accuracy and invariance.

The broader contribution is a systematic and reliable tool for *evaluating* and *selecting* invariant representations, enabling practitioners to build models that remain robust under distribution shift. DRIC fills the gap of a direct, environment-agnostic, normalized metric of expectation-level invariance compared to other existing metrics (Gretton et al., 2007; Zellinger et al., 2017). In Table 1 we provide a concise comparison table contrasting DRIC, MI/HSIC/MMD, domain classifiers, and risk-based criteria.

| Criterion | Measures | Uses $Y$ | Normalized |
|---|---|---|---|
| DRIC | Stability of $\mathbb{E}[Y \mid \phi(X), E]$ | Y | Y |
| Domain classifier | $\mathrm{MI}((\phi(X), Y), E)$ / env. predictability | optional | N |
| HSIC / MMD | Dependence / marginal shift of $\phi(X)$ | N | N |
| Risk variance / worst-case risk | Variation of loss across envs. | Y | N |

Table 1: Conceptual comparison of DRIC with existing criteria.

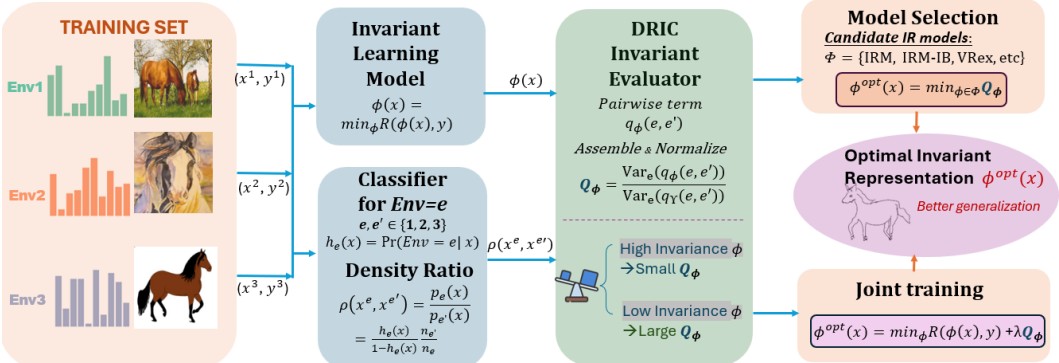

Figure 1: DRIC workflow for evaluating and selecting invariant representations.

## 2 BACKGROUND

### 2.1 INVARIANT RISK MINIMIZATION

We consider data drawn from environments $\mathcal{E}$, each inducing a distribution on $\mathcal{X} \times \mathcal{Y}$. Let $(X^e, Y^e)$ denote variables from $e \in \mathcal{E}$ and define the per-environment risk $R^e(f) = \mathbb{E}[\ell(f(X^e), Y^e)]$. Our goal is to control worst-case OOD risk using training environments $\mathcal{E}_{\mathrm{tr}} \subset \mathcal{E}$:

$$\min_{f:\mathcal{X}\to\mathcal{Y}} R^{\mathrm{OOD}}(f) = \min_f \max_{e \in \mathcal{E}} R^e(f).$$

This standard Empirical Risk Minimization (ERM) on pooled data may overfit environment-specific correlations in $\mathcal{E}_{\mathrm{tr}}$. To tackle this issue, Arjovsky et al. (2019) seek a representation that remains stable across different domains, while ensuring that stability in unseen environments as well, specifically finding an invariant predictor across $\mathcal{E}_{\mathrm{tr}}$ defined as follows.

**Definition 2.1.** Given an embedded space $\mathcal{H}$, a data representation $\phi : \mathcal{X} \to \mathcal{H}$ is said to be an **invariant representation** across environments $\mathcal{E}$ if there exists a classifier $w : \mathcal{H} \to \mathcal{Y}$ simultaneously optimal for all environments, i.e., for all $e \in \mathcal{E}$, $w \in \arg\max_{\bar{w}:\mathcal{H}\to\mathcal{Y}} R^e(\bar{w}, \phi)$. If the invariant representation $\phi$ elicits the classifier $w$, then $w \circ \phi$ is called an **invariant predictor**.

**Assumption 2.2** (Expectation-level invariance). If the optimal classifier in any environment of $\mathcal{E}$ can be written as a conditional expectation, then a data representation $\phi$ is invariant if and only if, for all $e, e' \in \mathcal{E}$ and all $h$ in the intersection of the supports of $\phi(X^e)$ and $\phi(X^{e'})$,

$$\mathbb{E}[Y^e|\phi(X^e) = h] = \mathbb{E}[Y^{e'}|\phi(X^{e'}) = h], \tag{1}$$

where $\mathbb{E}[Y^e|\phi(X^e)] = \mathbb{E}[Y \mid \phi(X), E = e]$ is the conditional expectation of $Y$ given $\phi(X)$ and $e$.

While some methods may adopt stronger conditions, the invariance on the conditional expectation remains a fundamental truth in all scenarios. We adopt Assumption 2.2 as a minimal and widely applicable invariance condition. By (1), *Invariant Risk Minimization (IRM)* can be posed as

$$\min_{\substack{\phi:\mathcal{X}\to\mathcal{H} \\ w:\mathcal{H}\to\mathcal{Y}}} \sum_{e\in\mathcal{E}_{\mathrm{tr}}} R^e(w, \phi), \quad \text{subject to } w \in \arg\min_{\bar{w}:\mathcal{H}\to\mathcal{Y}} \mathcal{J}(w, \phi), \text{ for all } e \in \mathcal{E}_{\mathrm{tr}}, \tag{2}$$

and is typically relaxed with a penalty $\mathcal{J}(w, \phi)$. *IRMv1* proposes $\mathcal{J}(w, \phi) = \|\nabla_w R^e(w, \phi)\|^2$ (Arjovsky et al., 2019). Under a stronger invariant assumption that $\mathbb{P}[Y|\phi(X^e)] = \mathbb{P}[Y|\phi(X^{e'})]$, Krueger et al. (2021) proposed *VREx* with $\mathcal{J}(w, \phi) = \mathrm{Var}(\mathcal{R}^e(w, \phi))$, while Chang et al. (2020) proposed *InvRat* with $\mathcal{J}(w, \phi) = \lambda(R_e(w, \phi) - R_e(w_e, \phi))$.

### 2.2 DISTRIBUTION SHIFT AND DENSITY RATIO

In statistics and machine learning, *distribution shift* refers to a mismatch between the training and test distributions (Masashi and Klaus-Robert, 2005). We extend this notion to multiple environments $\mathcal{E}$, each inducing a distribution on $\mathcal{X} \times \mathcal{Y}$; such shifts are common across sites, instruments, and time.

A traditional approach to address the difference between two distributions is to reweight the distribution by the density ratio to match the other. For example, for two closely-related variables $X_1, X_2 : \mathcal{X} \to \mathbb{R}$ with distributions $\mathbb{P}_1, \mathbb{P}_2$, respectively, observe that

$$\mathbb{E}_{X_1}(X_1) = \int_{\mathcal{X}} x \, d\mathbb{P}_1(x) = \int_{\mathcal{X}} x \, \frac{d\mathbb{P}_1(x)}{d\mathbb{P}_2(x)} \, d\mathbb{P}_2(x) = \mathbb{E}_{X_2}\left(X_2 \, \frac{d\mathbb{P}_1(X_2)}{d\mathbb{P}_2(X_2)}\right),$$

where $d\mathbb{P}_1(x)/d\mathbb{P}_2(x)$ denotes the likelihood ratio(**density ratio**), of $X_1$ over $X_2$. Thus, the density ratio can be applied to the data with distinct distributions across different environments.

Classical *covariate-shift* adaptation applies the same reweighting on $X$ to align source and target domains (Shimodaira, 2000; Sugiyama and Müller, 2005; Sugiyama et al., 2007; Quinonero-Candela et al., 2008; Reddi et al., 2015; Chen et al., 2016). We adopt this tool here and estimate $\rho_{e,e'}$ from data; robust and practical implementation details are provided in Appendix A.2.

## 3 METHODOLOGY

### 3.1 FROM INVARIANCE PROPERTY TO DRIC

Suppose that we obtain a data representation $\phi$ from (2) in IL methods, and we want to assess how close it is to an ideal invariant representation. Without loss of generality, we denote $Y \mid E = e$ by $Y^e$, $X \mid E = e$ by $X^e$, and $\mathbb{P}(x|E = e)$ by $\mathbb{P}^e(x)$. Although there exist stronger conditions for invariant representation shown in Section 5, we adopt the minimal assumption in 2.2, and thus our DRIC metric works for general IL methods. Then under assumption 2.2, for any $e, e' \in \mathcal{E}_{\mathrm{tr}}$, we have

$$
\mathbb{E}_{X^e} \left( \mathbb{E}(Y^e|\phi(X^e)) \right) = \int_{\mathcal{X}} \mathbb{E}\left(Y^e|\phi(X^e) = \phi(x)\right) d\mathbb{P}^e(x)
$$

$$
= \int_{\mathcal{X}} \mathbb{E}\left(Y^{e'}|\phi(X^{e'}) = \phi(x)\right) d\mathbb{P}^e(x) = \int_{\mathcal{X}} \mathbb{E}\left(Y^{e'}|\phi(X^{e'}) = \phi(x)\right) \frac{d\mathbb{P}^e(x)}{d\mathbb{P}^{e'}(x)} d\mathbb{P}^{e'}(x)
$$

$$
= \mathbb{E}_{X^{e'}} \left( \mathbb{E}(Y^{e'}|\phi(X^{e'}))\rho(X^e, X^{e'})(X^{e'}) \right), \tag{3}
$$

where $\rho(X^e, X^{e'})(x) := \mathrm{d}P_e/\mathrm{d}P_{e'}(x)$ is the point-wise density ratio of $X^e$ to $X^{e'}$ when they are continuous distributed. For any $e, e' \in \mathcal{E}_{\mathrm{tr}}$, let us define

$$
q_\phi(e, e') = \mathbb{E}_{X^{e'}} \left( \mathbb{E}(Y^{e'}|\phi(X^{e'}))\rho(X^e, X^{e'})(X^{e'}) \right). \tag{4}
$$

Clearly, $q_\phi(e, e) = \mathbb{E}_{X^e} \left( \mathbb{E}(Y^e|\phi(X^e)) \right)$, which can be denoted as $\mathbb{E}_{X|E=e} \left( \mathbb{E}(Y|\phi(X), E = e) \right)$. From (3), we observe that $q_\phi(e, e)$ differs from $q_\phi(e, e')$ only by a covariate-shift reweighting via the density ratio. Thus, (3) suggests that any $(q_\phi(e, e') - q_\phi(e, e))^2$ should be zero when $\phi$ is ideally invariant. We can then naturally argue that, for an arbitrary data representation $\phi$, the closer $(q_\phi(e, e') - q_\phi(e, e))^2$'s are to zero, the closer $\phi$ is to the ideal invariant representation.

In order to derive a robust quantity for the invariance of $\phi$ that is not affected by linear transformations of $Y$ or choices of $e, e'$, we propose a normalized quantity,

$$
Q_\phi = \frac{\sum_{e,e' \in \mathcal{E}_{\mathrm{tr}}} (q_\phi(e, e') - q_\phi(e, e))^2}{\sum_{e,e' \in \mathcal{E}_{\mathrm{tr}}} (q_\Upsilon(e, e') - q_\Upsilon(e, e))^2} = \frac{Var_E \{\mathbb{E}[Y|\phi(X), E] - \mathbb{E}(Y|E)\}}{Var_E \{\mathbb{E}[Y|\Upsilon(X), E] - \mathbb{E}(Y|E)\}} \tag{5}
$$

where the representation[1] $\Upsilon$ is the identity mapping defined by $\Upsilon(x) = x$. We introduce $R_\phi := Var_E \{\mathbb{E}[Y \mid \phi(X), E] - \mathbb{E}(Y \mid E)\}$ and then $Q_\phi = R_\phi / R_\Upsilon$, where $R_\Upsilon$ is the same variance computed for the identity representation $\Upsilon(X) = X$. DRIC value $Q_\phi$ measures how much the conditional expectation $\mathbb{E}[Y|\phi(X)]$ varies across environments, serving as a proxy for representation invariance. It has several desirable properties:

1. Environment-agnostic: Both the numerator and denominator of $Q_\phi$ average over all environment pairs, ensuring the score is not biased by specific environment choices.

2. Interpretable and normalized: The numerator captures the residual variation after applying $\phi$, while the denominator reflects the baseline variation without invariant learning. A value $Q_\phi < 1$ indicates that $\phi$ achieves some level of invariance.

3. Scale-invariant: $Q_\phi$ remains unchanged under linear transformations of the outcome $Y$, making it robust to rescaling.

We name this quantity $Q_\phi$ the **D**ensity-ratio-based **R**epresentation **I**nvariance **C**riterion (**DRIC**). A lower $Q_\phi$ value naturally indicates greater invariance of $\phi$. For very small values we report $\log Q_\phi$ for readability. Two representations have the same invariance level if $Q_\phi = Q_{\phi'}$.

*Remark* 3.1 (Sufficiency). It is natural that $Q_\phi = 0$ is a necessary condition for Assumption 2.2 to hold. We can verify that it is also sufficient almost surely. $Q_\phi = 0$ implies one of two cases:

---

[1]The predictor elicited by $\Upsilon$ equals to the predictor learned by ERM, and thus ERM DRIC is always 1.

- $\mathbb{E}(Y^{e'}|\phi(x)) = \mathbb{E}(Y^e|\phi(x))$ for all $x$, directly satisfying the desired invariance condition.
- The density ratio $\frac{dP^e(x)}{dP^{e'}(x)} = 0$, which occurs when the supports of $P^e$ and $P^{e'}$ are disjoint.

Case 1 confirms true invariance. Case 2, while also resulting in a zero difference, reflects disjoint covariate support, which makes invariant learning infeasible. In practice, this can be detected when estimated density ratios are close to zero across most inputs, indicating that the environments share no common input space, violating assumptions required for DRIC to function meaningfully.

*Remark* 3.2 (Invariance vs. Predictive Utility). IL methods target the conditional invariance $Y \perp E \mid \phi(X)$, while test accuracy is an indirect, distribution–dependent consequence. Accuracy conflates fit and shift magnitude, depends on the unknown test mix, and is non-monotone in invariance strength. We therefore measure invariance directly via DRIC $Q_\phi$ and use accuracy as a complementary metric to explore oracle representations, see Section 3.3 and 3.4.

## 3.2 Empirical estimation of DRIC

To estimate DRIC (5) using the training dataset $\mathcal{D}$, we start by estimating the important intermediate term $q_\phi(e, e')$ defined in (4). By Assumption 2.2, the true invariant representation $\phi$ and the elicited classifier $w$ leads to the conditional expectation of $Y$. Thus, in theory we can obtain population level expectation by,

$$q_\phi(e, e') = \mathbb{E}_{X^{e'}} \left( \frac{1}{n_{e'}} \sum_{i=1}^{n_{e'}} w \circ \phi(\mathbf{x}_i^{e'}) \rho(X^e, X^{e'})(\mathbf{x}_i^{e'}) \right).$$

Suppose that we empirically learn a classifier $\hat{w}$ and data representation $\hat{\phi}$ from $\mathcal{D}$. Intuitively, an estimator of $q_\phi(e, e')$ would consist of an empirical term,

$$\hat{q}_\phi(e, e') = \frac{1}{n_{e'}} \sum_{i=1}^{n_{e'}} \hat{w} \circ \hat{\phi}(\mathbf{x}_i^{e'}) \rho(X^e, X^{e'})(\mathbf{x}_i^{e'}), \quad \hat{q}_\phi(e, e) = \frac{1}{n_e} \sum_{i=1}^{n_e} \hat{w} \circ \hat{\phi}(\mathbf{x}_i^e). \tag{6}$$

With the general empirical estimator (6), all other terms appearing in DRIC value (5) can be naturally estimated. We list them as follows.

$$\hat{q}_\Upsilon(e, e) = \frac{1}{n_e} \sum_{i=1}^{n_e} \tilde{w}(\mathbf{x}_i^e), \quad \hat{q}_\Upsilon(e, e') = \frac{1}{n_{e'}} \sum_{i=1}^{n_{e'}} \tilde{w}(\mathbf{x}_i^{e'}) \rho(X^e, X^{e'})(\mathbf{x}_i^{e'}), \tag{7}$$

where $\tilde{w}$ is the baseline classifier learned without employing any additional representation. Using the empirical estimations (6) and (7), we can estimate DRIC in the following way,

$$\hat{Q}_\phi = \frac{\sum_{e,e' \in \mathcal{E}_{\mathrm{tr}}} (\hat{q}_\phi(e, e') - \hat{q}_\phi(e, e))^2}{\sum_{e,e' \in \mathcal{E}_{\mathrm{tr}}} (\hat{q}_\Upsilon(e, e') - \hat{q}_\Upsilon(e, e))^2}. \tag{8}$$

**Time and Memory Cost.** DRIC uses a classification-based method to estimate density ratios (Section A.2), making it both scalable and efficient. The computational complexity is $\mathcal{O}(ndT)$ in time and $\mathcal{O}(nd)$ in memory, where $n$ is the total number of samples, $d$ the input dimension, and $T$ the training steps. Runtime and memory usage scale linearly with data size, as detailed in Section A.4. This confirms DRIC's practical applicability with minimal overhead.

**Sample Complexity.** We can derive a sample complexity bound for estimating $q_\phi(e, e')$ using Hoeffding's inequality. To ensure an estimation error $\varepsilon$ with probability $1 - \alpha$, the required sample size must satisfy $n_{e'} \geq C\sigma^2 \log(\alpha^{-1})/\varepsilon^2$, where $\sigma^2$ is the variance of the weighted density ratio term. We show how stable DRIC estimates are achievable with sufficient samples in Section A.4.

### 3.3 Practical guidance of DRIC representation selection

In practice, DRIC can be a diagnostic tool for different representations $\phi$. It can guide the selection of feature extractors in domain generalization settings, where robustness is critical.

**Model selection.** For each candidate $\phi_h$ in pipelines $\Phi = \{\phi_h : h \in \mathcal{H}\}$, e.g., IRM/IB-IRM/VREx, we have DRIC scores $\widehat{Q}_{\phi_h}$ and validation risks $R_{\mathrm{val}}(\phi_h)$. Since $\widehat{Q}$ is environment-agnostic and normalized, $\{(R_{\mathrm{val}}(\phi_h), \widehat{Q}_{\phi_h})\}$ are comparable. Select $\phi^\star$ via any of:

(i) Accuracy-constrained: $\phi^\star = \arg\min_{\phi_h} \widehat{Q}_{\phi_h}$ s.t. $R_{\mathrm{val}}(\phi_h) \leq R_{\min} + \varepsilon$,

(ii) Target-invariance: $\phi^\star = \arg\min_{\phi_h} R_{\mathrm{val}}(\phi_h)$ s.t. $\widehat{Q}_{\phi_h} \leq Q^\star$,

(iii) Scalarization: $\phi^\star = \arg\min_{\phi_h} R_{\mathrm{val}}(\phi_h) + \lambda_{\mathrm{sel}} \widehat{Q}_{\phi_h}$,

where $R_{\min} = \min_h R_{\mathrm{val}}(\phi_h)$, $\varepsilon \geq 0$ is an accuracy tolerance, $Q^\star$ is a target invariance level. In practice, we can plot $\{(R_{\mathrm{val}}, \widehat{Q})\}$ and pick a Pareto knee.

**Joint training.** Instead of enumerating candidates, one can train $\phi$ by directly penalizing DRIC:

$$\min_{\phi, w} \quad R(\phi(X), Y; w) + \lambda \, Q_\phi, \tag{9}$$

where $R$ is the empirical risk and $Q_\phi$ is the DRIC objective. We implement (9) with alternating update $(\phi, w)$ to reduce $R + \lambda Q_\phi$ using environment-balanced mini-batches. As shown in Section 4.4, DRIC regularization yields more robust, domain-invariant representations with low $\widehat{Q}_\phi$ and strong OOD test accuracy. Theorem 3.5 shows the ideal lower bound for (9).

**Rule-of-thumb guidance.** Theoretically, $\widehat{Q}$ admits an information–theoretic lower bound even at ideal predictive risk (Section 3.4), DRIC = 0 is unattainable under when $Y$ and $E$ are dependent. *Safety/robustness/fairness–critical cases* prefer lower $\widehat{Q}$ stronger invariance and accept small accuracy drops. *Benchmark-style settings* prioritize accuracy, but use $\widehat{Q}$ to diagnose residual environment leakage and motivate robustness work. In Section 4.3, we show the fairness use case where we treat gender as environment.

### 3.4 Theoretical guarantee for DRIC

The accuracy guarantee of DRIC estimator Eq. (8) is ensured by the following theorem.

**Theorem 3.3.** *Suppose that, for any $e \in \mathcal{E}_{\mathrm{tr}}$, $\{\mathbf{x}_i^e\}_{i=1}^{n_e}$ are independently sampled from the distribution $\mathbb{P}^e$, and that there exists some $a, b > 0$ such that $an < n_e < bn$. Moreover, assume that $\phi$ is continuous, $w$ is Lipschitz continuous and $\sum_{e, e' \in \mathcal{E}_{\mathrm{tr}}} (q_\Upsilon(e, e') - q_\Upsilon(e, e))^2 > 0$. If $\|\hat{\phi} - \phi\|_\infty, \|\hat{w} - w\|_\infty$ converge to 0 and $\tilde{w}$ is uniformly convergent as $n \to \infty$, then $|\hat{Q}_\phi - Q_\phi| = o_p(1)$.*

This shows that the DRIC value closely approximates the true expectation (5). When $\hat{\phi}$ approaches the ideal invariant representation that captures all invariant features, the DRIC value approaches zero.

*Remark* 3.4. The within-environment independence assumption on $\{\mathbf{x}_i^e\}_{i=1}^{n_e}$ is not a necessary condition for the convergence of DRIC's estimation. In fact, any assumption that enables the Law of Large Numbers to hold is acceptable, such as the common scenario of stationary time series. For the sake of simplicity in the proof, we assumed independence for $\{\mathbf{x}_i^e\}_{i=1}^{n_e}$.

All the other assumptions stated in Theorem 3.3 are reasonable. The assumption on the bounds on $n_e$ guarantees that the sampling process is not imbalanced between different environments. For a practically applicable invariant representation, it is crucial that $\phi$ exhibits continuity and $w$ demonstrates Lipschitz continuity in a satisfactory manner. The necessity of the condition $\sum_{e, e' \in \mathcal{E}_{\mathrm{tr}}} (q_\Upsilon(e, e') - q_\Upsilon(e, e))^2 > 0$ arises from the fact that without it, there would be no need for invariant learning. Although the uniform convergence of $\hat{\phi}$ and $\hat{w}$ may depend on which approximation method used, it is satisfied for most of the existing methods in the literature. If the impact of the outliers is negligible, $\tilde{w}$ naturally exhibits uniform convergence.

According to Zhao et al. (2022), achieving a proper invariant representation involves balancing model accuracy and invariance. Specifically, for a given representation $\phi(X)$, we assume the loss function is the mean squared error, so the optimal empirical risk over parameters $w$ satisfies $\inf_w \mathbb{E}_\mathcal{D}\ell(w(\phi(X)), Y) = \mathbb{E}[\text{Var}(Y|\phi(X))]$. We now show an information bound on the DRIC value under the condition that $\phi(X)$ perfectly predicts $Y$, i.e. $\mathbb{E}[Var(Y \mid \phi(X))] = 0$.

**Theorem 3.5.** *Given $R_\phi$ for a representation $\phi$, suppose $\mathbb{E}[Var(Y|\phi(X))] = 0$ almost surely, which implies $Y$ is a deterministic function of $\phi(X)$. Then, under some regular conditions,*

$$\min_{\phi:\mathbb{E}[Var(Y|\phi(X))]=0} R_\phi \geq Cov^2(Y, E),$$

*where $Cov(Y, E)$ is the covariance between response $Y$ and the environment variable $E$.*

Specifically, we can have the lower bound for DRIC value is $Cov^2(Y, E)/R_\Upsilon$. This result indicates that if the representation $\phi(x)$ perfectly determines $Y$, i.e., $Y = f(\phi(x))$ almost surely, then the DRIC value cannot be made arbitrarily small. When $Y$ and $E$ are independent, the lower bound can be 0, meaning that the representation information among different environments is invariant. In other words, achieving ideal predictive accuracy imposes a nonzero lower bound on the DRIC measure, reflecting the inherent trade-off between invariance and exact prediction.

If $Y$ is highly environment-dependent, achieving both perfect prediction and high invariance is theoretically impossible. Therefore DRIC can help *diagnose overfitting* to environment-specific patterns, by quantifying residual environment dependence even when predictive performance is high.

## 4 Experiments

In this section, we present various empirical studies of our proposed DRIC metric. We first validate its effectiveness across three distinct settings: synthetic data ( 4.1), a domain generalization benchmark ( 4.2), and several real-world datasets ( 4.3). Besides, we show the utility of DRIC in joint optimization beyond post-hoc evaluation ( 4.4). Additional results are included in the Appendix, covering nonlinear settings of synthetic data (B.1), the DomainBed benchmark and ECE evaluation (B.2), group-invariant methods (B.4), robustness analysis (B.5), and model complexity (A.4).

### 4.1 Synthetic structural equation model data

We conduct experiments on a linear structural equation model (SEM) introduced by Arjovsky et al. (2019) and Krueger et al. (2021). We generate the experimental dataset from $(X^e, Y^e)$ for $e \in \{0.2, 2.0, 5.0\}$. $X^e = (X_1^e, X_2^e)$ contains a causal effect $X_1^e$ and a non-causal effect $X_2^e$, with both $X_1$ and $X_2$ being generated as 5-dimensional vectors. The generation details are as follows.

$$H^e \sim \mathcal{N}(0, e^2), \qquad\qquad X_1^e \sim W_{H\to 1}H^e + \mathcal{N}(0, e^2),$$
$$Y^e \sim W_{1\to Y}X_1^e + W_{H\to Y}H^e + \mathcal{N}(0, \sigma_y^2), \qquad X_2^e \sim W_{Y\to 2}Y + \mathcal{N}(0, \sigma_2^2).$$

In the context above, $W_{H\to 1}, W_{1\to Y}, W_{H\to Y}$ and $W_{Y\to 2}$ are all fixed parameters. The value of $W_{1\to Y}$ is consistently set to 1 in all experiments, while the settings of the remaining parameters are varied across different experimental scenarios. The following illustrations depict these variations.

Table 2: Summary of experimental settings combining hidden structure and noise types.

| No. | Hidden Paths | Noise Type | Notation | Description |
|-----|-------------|-----------|----------|-------------|
| 1 | Partially-observed (P) | Homoskedastic (O) | POU | $W \sim \mathcal{N}(0, 1); \sigma_y^2 = e^2, \sigma_2^2 = 1$ |
| 2 | Partially-observed (P) | Heteroskedastic (E) | PEU | $W \sim \mathcal{N}(0, 1); \sigma_y^2 = 1, \sigma_2^2 = e^2$ |
| 3 | Fully-observed (F) | Homoskedastic (O) | FOU | $W = 0; \sigma_y^2 = e^2, \sigma_2^2 = 1$ |
| 4 | Fully-observed (F) | Heteroskedastic (E) | FEU | $W = 0; \sigma_y^2 = 1, \sigma_2^2 = e^2$ |

We generate a total sample size of $n = 1300$ or $1800$ independent observations from the described structural equation modeling (SEM) model. We allocate 800 and 1200 samples, respectively, from the total samples to train the IRMv1, VREx, ERM, and LISA (Yao et al., 2022) models, and the remaining are test samples. Following the training phase using various methods, we obtain corresponding estimates for the representation $\hat{\phi}$ and the classifier $\hat{w}$ (or $\tilde{w}$ for ERM). These estimates are then used to calculate $\hat{Q}_\phi$. The entire process is repeated 10 times for each of the settings 1-4.

For clarity of illustration, we present our results using $\log_{10} \hat{Q}_\phi$ for both training and testing data in Figures 2. Firstly, The values of $\log_{10} \hat{Q}_\phi$ for IRMv1, VREx and LISA are consistently below 0 in all settings, demonstrating their superiority over ERM in terms of invariant learning. This result is reasonable since ERM does not capture the invariant patterns. Secondly, compared to IRMv1, VREx exhibits better invariance performance due to its stronger penalty on cross-environmental variance. We also observed greater variance in the DRIC value of IRMv1, suggesting potential instability of this method. Lastly, LISA achieves smaller DRIC values compared to VREx indicating the effectiveness of selective data augmentation in generating more invariant representations.

Moreover, we conduct further simulation studies under the nonlinear, non-Gaussian setting. We also calculate the $R^2$ in linear settings to show the explained variance in different methods and relate it with DRIC. In our synthetic data generation settings, the true value $q_\phi(e, e')$ is 0 in four settings. The results are shown in Figure 4 and Table 9, 19 in Appendix.

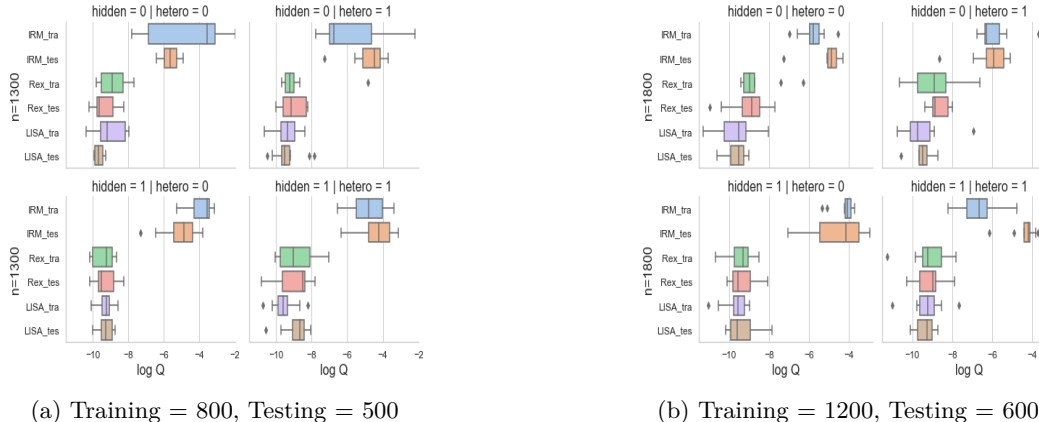

(a) Training = 800, Testing = 500  (b) Training = 1200, Testing = 600

Figure 2: DRIC values on the SEM task. The $y$-axis represents different methods: IRMv1, VREx, and LISA. The red line indicates the baseline $\log(\hat{Q}_\phi) = 0$ of ERM.

## 4.2 DomainBed

We evaluated DRIC on the DomainBed (Gulrajani and Lopez-Paz, 2020) benchmark using CMNIST dataset, which varies the level of spurious correlation across three environments (+90%, +80%, -90%). We tested invariant learning algorithms including IRM, VREx, GroupDRO, and IB-IRM, against the baseline ERM. The detailed experimental settings are included in Appendix B.1.

The results are presented in Figure 3a. Among the evaluated methods, the ERM baseline performed the worst, as it failed to learn invariant features and resulted in the highest DRIC score and lowest testing accuracy. In contrast, IB-IRM achieved the best performance with a testing accuracy of 50.60% and a DRIC score of 0.19. Other invariant learning algorithms performed between these two extremes. Ahuja et al. (2021) argue that their information-bottleneck term "enforces stronger invariance", and our empirical DRIC scores first quantify this claim at representation-level invariance in practice. This finding is also consistent with Yoshida and Naganuma (2024), further validates the effectiveness of DRIC as a metric for assessing model invariance.

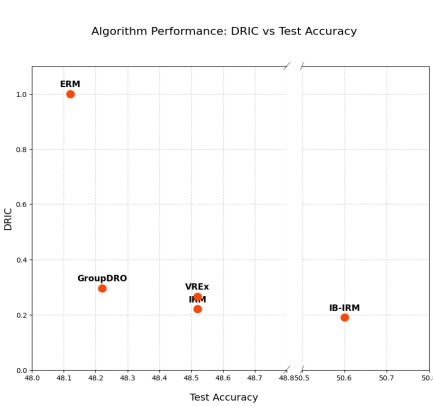
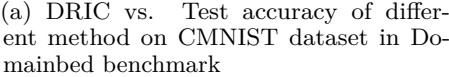
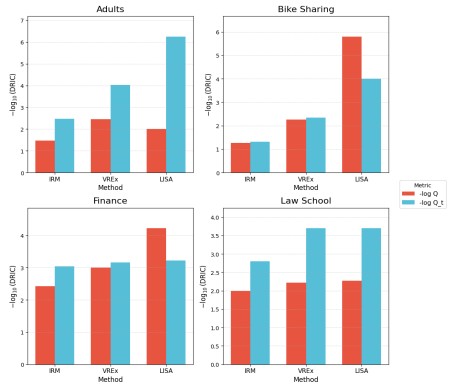

(a) DRIC vs. Test accuracy of different method on CMNIST dataset in Domainbed benchmark

(b) -log (DRIC) on training domain $(-\log_{10} \hat{Q})$ and on test domain $(-\log_{10} \hat{Q}_t)$ for Finance, Law School, Adult, and Bike Sharing datasets.

Figure 3: DRIC values on the Domainbed benchmark and real data analysis task.

## 4.3 REAL DATA

To evaluate the practical utility of DRIC, we conducted an analysis on four real-world datasets using three invariant learning methods (IRMv1, VREx, and LISA), benchmarked against a standard ERM model. The datasets include: the Finance Data, which uses U.S. stock market information to predict price variation, with each year from 2014 to 2018 serving as a distinct environment; the Law School Data, which predicts undergraduate GPA, where environments are constructed based on gender to simulate domain shifts; Adults Data, which predicts income levels, also using gender to define environments; and the Bike Sharing Data, which predicts rental counts, with each season treated as a separate environment. The details of implementation are shown in Appendix B.3.

The results of each dataset are presented in Figure 3b and Table **??**. Across different datasets, DRIC provides a consistent evaluation: with LISA consistently perform best (lowest DRIC) due to its selective augmentation technique, and IRMv1 demonstrated the weakest results (highest DRIC) due to its unstable regularization, which is aligned with previous literature (Yao et al., 2022). These results validate DRIC's utility as a robust metric for cross-dataset algorithm comparison.

## 4.4 DRIC AS A REGULARIZER FOR JOINT TRAINING

The previous sections have empirically validated DRIC as a robust post-hoc metric for evaluating the invariance of learned representations. Here we evaluate the joint optimization method described in Section 3.3 beyond post-hoc evaluation. Specifically, we compare DRIC-regularized model against ERM, VREx, and IRM, using experimental setup identical to Section 4.1. To ensure a fair comparison, the penalty hyperparameter $\lambda$, for each regularized method was chosen by cross-validation, with the optimal value selected based on performance on a held-out validation set.

As shown in Table 3, the DRIC-regularized model achieves the lowest Mean Squared Error (MSE) on both the training and testing domains. This superior performance suggests that using DRIC as a penalty provides a more effective optimization objective, guiding the model towards a more robust representation that achieves a better point on the Pareto frontier of accuracy and invariance.

Table 3: Compare DRIC-regularized model with ERM, VREx, and IRM on training and testing MSE .

| Method | MSE (Train) | MSE (Test) |
|---|---|---|
| ERM | 0.2616 | 0.8050 |
| VREx | 0.2775 | 0.7643 |
| IRM | 0.4334 | 0.7889 |
| **DRIC-regularized** | **0.2459** | **0.7472** |

## 5 RELATED WORK

Many follow-up works have been proposed to improve the performance of IRM. Rosenfeld et al. (2020) pointed out limitations of IRM in classification tasks. Wald et al. (2021) operate at the prediction level, typically using calibration error. Zhou et al. (2022) added a sparsity constraint to the network and trained a neural network that is sparse to prevent overfitting. Lin et al. (2022) extended *InvRat* using a Bayesian method with a posterior distribution of the classifier $w$. Chang et al. (2020), Koyama and Yamaguchi (2020), Ahuja et al. (2021), and Li et al. (2022) considered the invariant learning problem from the perspective of information theory. Mahajan et al. (2021) introduced a novel regularizer to match the representation of the same object in different environments. Creager et al. (2021) proposed *EIIL*, which attempts to automatically partition a dataset into different environments to learn environment labels that maximize the IRM's penalty. Wang et al. (2022) proposed a simple post-processing method for solving the IRM problem without retraining the model. Zhang et al. (2023a) proposed *Generalization Adjustment* to address scenarios where the support of multi-domain data is not available during mini-batch training. Yang et al. (2023) addressed the limitations of existing methods in handling sufficiency and necessity properties in out-of-distribution generalization by introducing the sufficiency and necessity causes risk. Huang et al. (2024) proposed the EVIL algorithm, which utilizes distribution knowledge to identify parameters sensitive to distribution shifts. Zhang et al. (2023b) introduced a metric that quantifies the presence of covariate shift. Testing for distributional invariance under arbitrary symmetry groups is another direction of invariance evaluation (Chen et al., 2022). Soleymani et al. (2025) introduced a robust, kernel-based framework for testing group invariance in data and support for subtle asymmetries. Koning and Hemerik (2024) proposed a more efficient invariance testing method by replacing random transformation subsets with fixed subgroups. Chiu and Bloem-Reddy (2023) developed general non-parametric tests for distributional equivariance under group actions using kernel methods. Nguyen et al. (2024); Kim et al. (2025) develops recent SOTA for domain generalization beyond representation learning. In contrast, DRIC directly measures representation-level expectation invariance and is normalized and environment-agnostic, making scores comparable across IL methods and datasets.

## 6 CONCLUDING REMARKS

This paper proposed DRIC, a density ratio-based criterion to assess the domain-invariant representations. DRIC relies on density–ratio estimation and well-specified environments, and misspecified or imbalanced environments can bias $\widehat{Q}_\phi$. In the future, we will calibrate $\widehat{Q}_\phi$ with uncertainty quantification, develop estimators robust to imbalance or weak overlap, and refine environments, while strengthening finite-sample guarantees.

DRIC offers a pathway toward more trustworthy and fair AI systems, enabling models that perform consistently across different environments and populations. From robust healthcare predictions across hospitals to reliable autonomous driving under varying conditions, DRIC can help mitigate algorithmic bias and promote equitable AI deployment in real-world applications.

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

Acknowledgment: We use large language models (LLM) to aid or polish writing. The authors drafted the manuscript; LLM was used only to correct grammar and typographical errors. The authors reviewed all edits and are responsible for the content.

## A  Technical Appendices and Supplementary Material

### A.1  Theoretical Proofs

*Proof of Theorem 3.3.* We begin by proving $|\hat{q}_\phi(e, e') - q_\phi(e, e')| = o_p(1)$ for any $e, e' \in \mathcal{E}_{\text{tr}}$. For ease of illustration, we denote that

$$\tilde{q}_\phi(e, e') = \frac{1}{n_{e'}} \sum_{i=1}^{n_{e'}} w \circ \phi(\mathbf{x}_i^{e'}) \rho(X^e, X^{e'})(\mathbf{x}_i^{e'}).$$

Since $||\hat{\phi} - \phi||_\infty, ||\hat{w} - w||_\infty = o(1)$, $\phi$ is continuous and $w$ is Lipschitz continuous, we clearly have $||\hat{w} \circ \hat{\phi} - w \circ \phi||_\infty = o(1)$. Thus, we can derive that

$$
\begin{aligned}
|\hat{q}_\phi(e, e') - \tilde{q}_\phi(e, e')| &= \left| \frac{1}{n_{e'}} \sum_{i=1}^{n_{e'}} \hat{w} \circ \hat{\phi}(\mathbf{x}_i^{e'}) \rho(X^e, X^{e'})(\mathbf{x}_i^{e'}) - \frac{1}{n_{e'}} \sum_{i=1}^{n_{e'}} w \circ \phi(\mathbf{x}_i^{e'}) \rho(X^e, X^{e'})(\mathbf{x}_i^{e'}) \right| \\
&= \left| \frac{1}{n_{e'}} \sum_{i=1}^{n_{e'}} \left( \hat{w} \circ \hat{\phi}(\mathbf{x}_i^{e'}) - w \circ \phi(\mathbf{x}_i^{e'}) \right) \rho(X^e, X^{e'})(\mathbf{x}_i^{e'}) \right| \\
&\leq ||\hat{w} \circ \hat{\phi} - w \circ \phi||_\infty \cdot \left| \frac{1}{n_{e'}} \sum_{i=1}^{n_{e'}} \rho(X^e, X^{e'})(\mathbf{x}_i^{e'}) \right|.
\end{aligned}
$$

Since $n_{e'} > an$, by Law of Large Numbers, we have

$$\frac{1}{n_{e'}} \sum_{i=1}^{n_{e'}} \rho(X^e, X^{e'})(\mathbf{x}_i^{e'}) \overset{a.s.}{\to} \mathbb{E}_{X^{e'}} \left( \rho(X^e, X^{e'})(\mathbf{x}_i^{e'}) \right) = \int_{\mathcal{X}} \frac{d\mathbb{P}^e(x)}{d\mathbb{P}^{e'}(x)} d\mathbb{P}^{e'}(x) = 1.$$

Therefore, with probability 1 we have

$$|\hat{q}_\phi(e, e') - \tilde{q}_\phi(e, e')| \leq ||\hat{w} \circ \hat{\phi} - w \circ \phi||_\infty \cdot 1 = o(1).$$

On the other hand, again by Law of Large Numbers, we have

$$\tilde{q}_\phi(e, e') = \frac{1}{n_{e'}} \sum_{i=1}^{n_{e'}} w \circ \phi(\mathbf{x}_i^{e'}) \rho(X^e, X^{e'})(\mathbf{x}_i^{e'}) \overset{a.s.}{\to} \mathbb{E}_{X^{e'}} \left( \mathbb{E}(Y^{e'} | \phi(X^{e'})) \rho(X^e, X^{e'})(X^{e'}) \right) = q_\phi(e, e').$$

Hence, it follows that

$$|\hat{q}_\phi(e, e') - q_\phi(e, e')| \leq |\hat{q}_\phi(e, e') - \tilde{q}_\phi(e, e')| + |\tilde{q}_\phi(e, e') - q_\phi(e, e')| = o_p(1).$$

Next, we note that $|\hat{q}_\phi(e, e) - q_\phi(e, e)| = o_p(1)$ is a direct conclusion of the first step. Thus, we clearly have

$$\left| \sum_{e, e' \in \mathcal{E}_{\text{tr}}} (\hat{q}_\phi(e, e') - \hat{q}_\phi(e, e))^2 - \sum_{e, e' \in \mathcal{E}_{\text{tr}}} (q_\phi(e, e') - q_\phi(e, e))^2 \right| = o_p(1).$$

Following similar procedures, we can show that

$$\left| \sum_{e, e' \in \mathcal{E}_{\text{tr}}} (\hat{q}_\Upsilon(e, e') - \hat{q}_\Upsilon(e, e))^2 - \sum_{e, e' \in \mathcal{E}_{\text{tr}}} (q_\Upsilon(e, e') - q_\Upsilon(e, e))^2 \right| = o_p(1).$$

Therefore, since $\sum_{e, e' \in \mathcal{E}_{\text{tr}}} (q_\Upsilon(e, e') - q_\Upsilon(e, e))^2 > 0$, it follows that $|\hat{Q}_\phi - Q_\phi| = o_p(1)$. ∎

*Proof of Theorem 3.5.* The optimal solution to the loss function $\inf_\phi \mathbb{E}_\mathcal{D} \ell(y, \phi(x))$ is $\phi(x) = \mathbb{E}[\mathrm{Var}(y|\phi(x))]$. Meanwhile, the DRIC value can be written as

$$\mathrm{Var}_E \left\{ \mathbb{E}[y|\phi(x), E] - \mathbb{E}(Y|E) \right\}.$$

We denote $\phi(X)$ by $Z$, and denote the difference of conditional means by $\Delta(Z, E) = \mathbb{E}[Y \mid Z, E] - \mathbb{E}[Y \mid E]$. Thus the $R_\phi$ is $Var(\Delta(Z, E))$ over the joint distribution of $(Z, E)$.

By law of total variance, we have $Cov(Y, E) = \mathbb{E}\left[Cov(Y, E|Z)\right] + Cov(\mathbb{E}(Y|Z, \mathbb{E}(E|\phi(x))))$. The optimal risk minimizer constraint $Var(\mathbb{E}[Y \mid Z]) = Var(Y)$ means that all the variance of $Y$ is explained by $Z$. Equivalently, we can write $Y = f(Z) + \varepsilon$, with $\varepsilon \perp Z$ and $Var(\varepsilon) = 0$. Thus, $Y$ is effectively a deterministic function of $Z$. In other words, $Y = f(Z)$ almost surely. Given $Y = f(Z) + \epsilon$, we have $\mathbb{E}[Y \mid Z, E] = \mathbb{E}[f(Z) \mid Z, E] = f(Z)$ for any $Z$.

Since $Y$ is almost surely determined by $Z$. Hence, $\Delta(Z, E) = f(Z) - \mathbb{E}[Y \mid E]$. Thus,

$$\begin{aligned}
Var(\Delta(Z, E)) &= Var(f(Z) - \mathbb{E}[Y \mid E]) \\
&= \mathbb{E}\left\{ f(Z) - \mathbb{E}(Y \mid E) - \mathbb{E}\left[ f(Z) - \mathbb{E}(Y \mid E) \right] \right\}^2 \\
&= Var\left\{ \mathbb{E}[Y - \mathbb{E}(Y|E)|Z] \right\} + \mathbb{E}Var\{ \mathbb{E}[Y - \mathbb{E}(Y|E)|Z] \} \\
&\geq Var\{ \mathbb{E}(Y|Z) - \mathbb{E}(Y|E, Z) \} + \mathbb{E}\,\mathrm{Var}(\mathbb{E}(Y|E)|Z) - \mathbb{E}\,\mathrm{Var}(Y|Z) \\
&= \mathrm{Var}(Y) + \mathrm{Var}(\mathbb{E}(Y|E)) - Cov(Y, \mathbb{E}(Y|E)) \\
&\geq \mathrm{Var}(\mathbb{E}(Y|E))
\end{aligned}$$

By Cauchy-Schwarz inequality, we have $Cov(Y, \mathbb{E}(Y \mid E))^2 \leq Var(Y)Var(\mathbb{E}(Y \mid E))$. By law of total variance, we have

$$Var(\mathbb{E}(Y \mid Z)) = Var(\mathbb{E}[\mathbb{E}(Y \mid Z) \mid E]) + \mathbb{E}[Var(\mathbb{E}(Y \mid Z) \mid E)]$$

Rearranging, we get:

$$\begin{aligned}
Var(\mathbb{E}(Y \mid E)) &= Var(\mathbb{E}(Y \mid Z)) - \mathbb{E}[Var(\mathbb{E}(Y \mid Z, E))] \\
&= \mathrm{Var}(Y) + \mathrm{Var}(\mathbb{E}(Y|Z, E)).
\end{aligned}$$

Suppose that $\mathbb{E}(Y \mid Z)$ is a convex function of $Z$, then by Jensen's inequality, we have

$$\mathbb{E}(Y \mid E) = \mathbb{E}[\mathbb{E}(Y \mid Z) \mid E] \geq \mathbb{E}(Y \mid E(Z \mid E))$$

We continue to use law of total variance, thus we will have

$$\begin{aligned}
Var(\mathbb{E}(Y \mid E)) &\geq Var\{ \mathbb{E}(Y \mid E(Z \mid E)) \} \\
&= \mathrm{Var}(Y) - \mathbb{E}\,\mathrm{Var}(Y \mid E(Z \mid E)) \\
&= Var[\mathbb{E}(Y|\mathbb{E}(Z|E))] \\
&\geq Var f[\mathbb{E}(Z|E(Z, E))] \\
&\geq Var[f(Z)E(Z, E)] \geq \mathrm{Var}(Y)\,\mathrm{Var}(\mathbb{E}(Z \mid E)).
\end{aligned}$$

From Zhao et al. (2022) we have $\mathrm{Var}(\mathbb{E}(E|Z)) \geq \mathrm{Var}(E)\rho_{YE}^2$, where $\rho_{YE}$ is the correlation coefficient of $Y$ and $E$.

Therefore, the DRIC value is bounded by

$$Var(\Delta(Z, E)) \geq \mathrm{Var}(Y)\,\mathrm{Var}(E)\rho_{YE}^2 = Cov^2(Y, E).$$

∎

## A.2 Estimation of the density ratio

In the main paper, we estimated DRIC using the learned $\hat{w}$ and $\hat{\phi}$, under the assumption of known likelihood ratio $\rho(X^e, X^{e'})$. Under certain assumptions regarding the distributions of $X^e$, such as the normal distribution, the likelihood functions of $X^e$ can be easily estimated by calculating the sample means and variances from $\{\mathbf{x}_i^e\}_{i=1}^{n_e}$. However, in a more general scenario, we do not hold any prior knowledge about $\mathbb{P}_e$, which necessitates estimating $\rho(X^e, X^{e'})$ without assuming any specific data distributions. Therefore, in this subsection,

we present a generic estimation strategy for the likelihood ratio, which extends the application of DRIC to a broader context.

Our method mainly follows the work done by Tibshirani et al. (2019). For two environments $e_1, e_2 \in \mathcal{E}_{\text{tr}}$, let $(X_{e_1,e_2}, E)$ be a pair of variables that is identically distributed as $(X^e, e)|e = e_1$ or $e_2$. Tibshirani et al. (2019) showed that

$$\frac{\Pr(E = e_1 | X_{e_1,e_2} = x)}{\Pr(E = e_2 | X_{e_1,e_2} = x)} = \frac{\Pr(E = e_1)}{\Pr(E = e_2)} \frac{d\mathbb{P}^{e_1}(x)}{d\mathbb{P}^{e_2}(x)}.$$

By observing that $\Pr(E = e_1 | X_{e_1,e_2} = x) + \Pr(E = e_2 | X_{e_1,e_2} = x) = 1$, we have

$$\frac{d\mathbb{P}^{e_1}(x)}{d\mathbb{P}^{e_2}(x)} = \frac{\Pr(E = e_1 | X_{e_1,e_2} = x)}{1 - \Pr(E = e_1 | X_{e_1,e_2} = x)} \cdot \frac{\Pr(E = e_2)}{\Pr(E = e_1)}.$$

Apparently, $\Pr(E = e_2)/\Pr(E = e_1)$ can be estimated by $n_{e_2}/n_{e_1}$. On the other hand, we consider the dataset denoted by $\mathcal{C}e_1, e_2$, defined as follows:

$$\mathcal{C}e_1, e_2 = (\mathbf{x}_i^e, e) : e = e_1 \text{ or } e_2, 1 \le i \le n_e.$$

We can then employ various classifiers, such as logistic regression or random forest, to estimate the conditional probability of class membership for $\mathcal{C}_{e_1,e_2}$. Subsequently, if $\hat{p}(x)$ represents the classifier's estimate of $\Pr(E = e_1 | X_{e_1,e_2} = x)$, we can then estimate $\rho(X^{e_1}, X^{e_2})(x) = d\mathbb{P}^{e_1}(x)/d\mathbb{P}^{e_2}(x)$ by

$$\hat{\rho}(X^{e_1}, X^{e_2})(x) = \frac{n_{e_2}\hat{p}(x)}{n_{e_1}(1 - \hat{p}(x))}. \tag{10}$$

Density ratio estimation is a well-studied area, and the method described here belongs to a general class of probabilistic classification approaches. Two other classes of density ratio estimation methods include moment matching and the minimization of $f$-divergences, such as the Kullback-Leibler divergence. For a comprehensive review of these approaches and the underlying theory, we refer readers to the work of Sugiyama et al. (2012).

*Remark* A.1. Similar to Cai et al. (2023), we employs the density ratio of covariates as a means to quantify the distribution shifts encountered across different environments. Our approach does not require a strict covariate shift assumption that $P_{Y^{e'}|X^{e'}} = P_{Y^e|X^e}$. In fact, our method is flexible and can be applied to various distribution shift scenarios.

We further evaluate density ratio estimation across dimensions 1, 5, and 10 using MLP, RuLSIF, and Logistic Regression. The source distribution is $P \sim \mathcal{N}(0, I)$ and the target distribution is $Q \sim \mathcal{N}(0.2, 1.5I)$, with 10,000 samples per setting. Performance is measured by the Mean Absolute Error (MAE) of log density ratios. As shown in Table 4 below, classification-based methods (MLP and Logistic Regression) strike a favorable balance between accuracy and computational efficiency.

Table 4: Comparison of different density ratio estimators.

| Dimension | MLP | RuLSIF | Logistic |
|---|---|---|---|
| 1 | 0.0894 | 0.4899 | 0.2029 |
| 5 | 0.4540 | 1.4500 | 0.5302 |
| 10 | 0.9004 | 2.5778 | 0.8002 |

### A.3 Connection to Representation level Density-Ratio

In practice, we can compute the density ratio in representation space $Z$ where $Z = \phi(x)$. For any $e, e' \in \mathcal{E}_{\text{tr}}$, we have

$$\mathbb{E}_{Z^e}\left(\mathbb{E}(Y^e | Z^e)\right) = \mathbb{E}_{Z^{e'}}\left(\mathbb{E}(Y^{e'} | Z^{e'})\rho(Z^e, Z^{e'})(Z^{e'})\right), \tag{11}$$

where $\rho(Z^e, Z^{e'})(z) := dP_e/dP_{e'}(z)$ is the point-wise density ratio of $Z^e$ to $Z^{e'}$ when they are continuous distributed. Moreover, this is a special case for any $Z$-measurable function $g$ that

$$\mathbb{E}_{X^e}\left(g(X^e)\right) = \mathbb{E}_{Z^e}\left(g(Z^e)\right) = \mathbb{E}_{Z^{e'}}\left(g(Z^{e'})\rho(Z^e, Z^{e'})\right),$$

when $\phi$ mapping from $X$ to $Z$ is (i) differentiable and (ii) invertible. If $Z$ is the ideal domain-invariant feature such that $Y \perp E \mid Z$, then $\mathbb{E}(\rho(Z^e, Z^{e'})) = 1$. The equation will hold that when our density ratio and the conditional expectation are calculated in the same representation space. In practice, we can compute ratios on the same representation we evaluate when the representation $\phi$ satisfies (i) and (ii). However, most of the neural network-based algorithms do not satisfy these conditions.

Instead, we demonstrate that under the stronger assumption of Invariance (i.e., $\mathcal{R}^e(f) = \mathcal{R}^{e'}(f)$, Krueger et al. (2021)), the calculation of DRIC is theoretically grounded in the representation space $\phi(x)$ rather than the raw input space $X$. This formulation naturally circumvents issues related to weak overlap or high dimensionality in $X$.

We start with the definition of Risk for an environment $e$:

$$\mathcal{R}^e(f) = \mathbb{E}_{(x^e, y^e) \sim p_e(X,Y)}[l(f(x^e), y^e)] = \int_x \int_y l(f(x), y) \cdot p_e(x, y)\, dx\, dy \qquad (12)$$

We introduce the following assumptions:

- **Assumption 1: Feature Disentanglement**: $x = g(x_{\text{inv}}, x_{\text{spu}})$.
- **Assumption 2: Spurious-Free Predictor**: $f(x) = f(x_{\text{inv}})$.
- **Assumption 3: Invariant Causal Mechanism**: $Y \perp X_{\text{spu}}|X_{\text{inv}}$, and $p_e(y|x_{\text{inv}}, x_{\text{spu}}) = p(y|x_{\text{inv}})$ ($p(y|x_{\text{inv}})$ is invariant across environments).

Based on these assumptions, we expand the risk $\mathcal{R}^e(f)$:

$$\mathcal{R}^e(f) = \int_x \int_y l(f(x), y) \cdot p_e(x, y)\, dx\, dy$$

$$= \int_{x_{\text{inv}}} \int_{x_{\text{spu}}} \int_y l(f(x_{\text{inv}}), y) \cdot p_e(x_{\text{inv}}, x_{\text{spu}}, y)\, dy\, dx_{\text{spu}}\, dx_{\text{inv}}$$

$$= \int_{x_{\text{inv}}} \int_{x_{\text{spu}}} \int_y l(f(x_{\text{inv}}), y) \cdot p_e(y|x_{\text{inv}}, x_{\text{spu}}) \cdot p_e(x_{\text{inv}}, x_{\text{spu}})\, dy\, dx_{\text{spu}}\, dx_{\text{inv}}$$

$$= \int_{x_{\text{inv}}} \int_y l(f(x_{\text{inv}}), y) \cdot p(y|x_{\text{inv}}) \cdot p_e(x_{\text{inv}})\, dy\, dx_{\text{inv}}$$

Consistent with our main text, we denote the invariant representation $x_{\text{inv}}$ as $\phi(x)$. Consequently, the risk for environment $e$ is reformulated as:

The risk for environment $e$ can be written as:

$$\mathcal{R}^e(f) = \int_{y^e} \int_{\phi(x^e)} l(f(\phi(x^e)), y^e) \cdot p(y^e|\phi(x^e)) \cdot p_e(\phi(x^e))\, d\phi(x^e)\, dy^e \qquad (13)$$

Similarly, the risk for environment $e'$ is

$$\mathcal{R}^{e'}(f) = \int_{y^{e'}} \int_{\phi(x^{e'})} l(f(\phi(x^{e'})), y^{e'}) \cdot p(y^{e'}|\phi(x^{e'})) \cdot p_{e'}(\phi(x^{e'}))\, d\phi(x^{e'})\, dy^{e'} \qquad (14)$$

Finally, we bridge the risk formulations of the two environments by introducing the density ratio in the representation space:

$$\mathcal{R}^e(f) = \int_y \int_{\phi(x)} l(f(\phi(x)), y) \cdot p(y|\phi(x)) \cdot p_e(\phi(x))\, d\phi(x)\, dy$$

$$= \int_y \int_{\phi(x)} l(f(\phi(x)), y) \cdot p(y|\phi(x)) \cdot \left(\frac{p_e(\phi(x))}{p_{e'}(\phi(x))}\right) \cdot p_{e'}(\phi(x))\, d\phi(x)\, dy$$

This derivation proves that the DRIC under stronger invariance assumption relies only on the density ratio of the representations:

$$\rho(\phi) = \frac{p_e(\phi(x))}{p_{e'}(\phi(x))}$$

This theoretical framing ensures robustness:

Shared Support: Even if the raw inputs $X$ have disjoint support (due to environment-specific spurious features), the criterion remains valid because the learned invariant features $\phi(x)$ share support by definition.

Computational Efficiency: Estimating the density ratio in the low-dimensional latent space $\phi(x)$ is significantly more stable and efficient than in the high-dimensional input space $X$.

### A.4 Sample Requirements and Computational Complexity of DRIC Estimation

The computational overhead primarily arises from estimating the density ratio, which is similar in form to what is used in KL divergence estimation. We adopt a classification-based approach to estimate this ratio (as detailed in Appendix Section 3), which is both practical and scalable.

Theoretically, the overall time complexity of DRIC is $\mathcal{O}(ndT)$ where $n$ is the total number of samples across environments, $d$ is the input dimension and $T$ is the number of optimization iterations for training the density ratio estimator. The memory complexity is $\mathcal{O}(nd)$. which accounts for storing all samples and their corresponding representations.

Runtime and memory usage for DRIC across different datasets are reported in Table 2 and 3 in the additional experiment. We observe that with a 3-layer MLP classifier, the computation time increases approximately linearly with $n_{\text{train}}^e$, and computing DRIC with $n_{\text{train}}^e = 700$ takes approximately 1 second in total, which indicates the minimal computational requirement for DRIC estimation.

While we do not derive an exact theoretical lower bound in the main paper, we agree that characterizing the sample complexity of the DRIC estimator is critical. Below, we provide a simple theoretical justification based on concentration inequalities, and complement it with empirical observations.

Let $\sigma^2 = Var[\omega\phi(x_i^{e'})\rho(X^e, X^{e'})(x^{e'})]$. Since $\hat{q}_\phi(e, e')$ is computed as the empirical average of $n_{e'}$ i.i.d. terms, we can obtain a general concentration bound via Hoeffding's inequality:

$$\mathbb{P}\{|\hat{q}_\phi(e, e') - \mathbb{E}_{x^{e'}}(\hat{q}_\phi(e, e'))| \geq \varepsilon\} \leq \exp\left(\frac{-Cn_{e'}\varepsilon^2}{\sigma^2}\right),$$

for some constant $C$. Rearranging gives the sample size $n_{e'}$ needed to bound the estimation error by estimation error $\varepsilon$ with probability $1 - \alpha$, $n_{e'} \geq C\sigma^2 \log(\alpha^{-1})/\varepsilon^2$. This suggests that the required number of samples grows proportionally with the variance of the weighted density ratio term and inversely with the square of the target accuracy $\varepsilon$. In our experiments, we observe that the DRIC estimate stabilizes with as few as 300 samples per environment, depending on the dimensionality and distributional divergence.

Now we investigate the sample requirements and computational complexity of DRIC estimation under different subsample size. To estimate the sample requirement for DRIC calculation, we analyze the relationship between subsample size and the classification accuracy of the MLP classifier. To estimate the time consumption of DRIC calculation, we record the total computation time for DRIC each run.

The data is generated following the procedure outlined in the Section 1. We first train ERM and IRM models on the the full training set with $n_{\text{train}}^e = 1000$ for each environment. We then fix the trained models and compute the DRIC estimates on sub-sampled training sets with $n_{\text{sub}}^e = [100, 300, 500, 700, 900]$ per environment. The DRIC value, time consumption and classification accuracy are presented in Table 5 and 6.

We observe that that the MLP classifier achieves a classification accuracy of 0.8 when using DRIC estimated from 300 samples per environment, for both ERM and IRM. This indicates that $n_{\text{train}}^e = 300$ is sufficient for accurate DRIC estimation under the current setting. We also observe that the computation time increases approximately linearly with with $n_{\text{sub}}^e$, and computing DRIC with $n_{\text{sub}}^e = 700$ takes approximately 1 second in total, which indicates the minimal computational requirement for DRIC estimation.

Table 5: Results for ERM

| subsample | DRIC | DRIC TIME | train loss | classification accuracy |
|---|---|---|---|---|
| 100 | 0.0092 | 0.1095 | 0.5142 | 0.7417 |
| 300 | 0.0233 | 0.1973 | 0.5088 | 0.8194 |
| 500 | 0.0119 | 0.6707 | 0.5272 | 0.9183 |
| 700 | 0.0126 | 0.9883 | 0.5369 | 0.9143 |
| 900 | 0.0118 | 1.1677 | 0.5393 | 0.9269 |

Table 6: Results for IRM

| subsample | DRIC | DRIC TIME | train loss | classification accuracy |
|---|---|---|---|---|
| 100 | 0.0212 | 0.1125 | 0.5746 | 0.7583 |
| 300 | 0.0053 | 0.3961 | 0.5190 | 0.8806 |
| 500 | 0.0084 | 0.6134 | 0.5290 | 0.8733 |
| 700 | 0.0051 | 1.2258 | 0.5118 | 0.9417 |
| 900 | 0.0053 | 1.1660 | 0.5225 | 0.9269 |

## B  ADDITIONAL NUMERICAL RESULTS

### B.1  SYNTHETIC DATA EXPERIMENT

#### B.1.1  EXPERIMENTAL SETTINGS

For synthetic data experiment, the samples are generated from three environments e = 0.2, 2, 5. For IRM and VREx, the invariance regularizer lambda is cross validated using the environment = 5. The experimental settings are listed in Table 7.

Table 7: Settings for Synthetic Data

| Parameter | Value |
|---|---|
| Lambda | 0, 1e-4, 1e-1 |
| Iterations of penalty annealing | 1 |
| Number of repetitions | 5 |
| L2 regularizer | 1 |
| Learning rate | 1e-3 |
| Number of iterations | 20 |
| Number of training samples | 800, 1200 |
| Number of testing samples | 500, 600 |

Table 8: MLP architecture

| # | Layer |
|---|-------|
| 1 | Linear (in_features=input_size, out_features=256, bias=True) |
| 2 | ReLU |
| 3 | Dropout (p=0.5) |
| 4 | Linear (in_features=256, out_features=256, bias=True) |
| 5 | ReLU |
| 6 | Dropout (p=0.5) |
| 7 | Linear (in_features=256, out_features=1, bias=True) |

### B.1.2 ADDITIONAL RESULTS FOR SYNTHETIC DATA: LINEAR SETTING

Under the setting shown above, we obtain the results shown in Table 9.

Table 9: The estimated $\log_{10} \hat{Q}_\phi$ under different settings where true $q_\phi = 0$, the sample size is 1300(above) and 1800(below) respectively.

| Method | IRMv1 | VREx | LISA |
|--------|-------|------|------|
| POU | $-4.493(1.173)$ | $-9.292(0.456)$ | $-9.344(0.175)$ |
| FOU | $-5.231(2.920)$ | $-9.142(0.556)$ | $-9.468(0.938)$ |
| PEU | $-4.603(1.134)$ | $-8.970(0.923)$ | $-9.347(0.604)$ |
| FEU | $-5.241(2.553)$ | $-8.915(1.202)$ | $-9.476(0.302)$ |

| Method | IRMv1 | VREx | LISA |
|--------|-------|------|------|
| POU | $-4.398(1.155)$ | $-9.462(0.661)$ | $-9.534(0.473)$ |
| FOU | $-5.400(0.707)$ | $-8.835(1.009)$ | $-9.761(1.022)$ |
| PEU | $-4.321(1.134)$ | $-9.380(0.679)$ | $-9.520(0.344)$ |
| FEU | $-5.886(1.508)$ | $-8.886(1.466)$ | $-9.607(0.944)$ |

### B.1.3 ADDITIONAL RESULTS FOR SYNTHETIC DATA: NONLINEAR SETTING

We conduct additional simulation studies under nonlinear and non-Gaussian setting with modified original synthetic structural equation model. We generate the data $(X^e, Y^e)$ for environment $e \in \{0.1, 1.0, 10.0\}$. $X^e = (X_1^e, X_2^e)$ contains a causal effect $X_1^e$ and a non-causal effect $X_2^e$, and both $X_1^e$ and $X_2^e$ are generated as 1-dimensional normal vectors. Specifically, we employ a quadratic relationship between $X_1$ and $Y$ and replace Gaussian noise with a noise following Student T distribution with $df = 3$. The data generation settings are detailed as below:

$$H^e \sim \mathcal{N}(0, e^2),$$
$$X_1^e \sim W_{H \to 1} H^e + \mathcal{N}(0, e^2),$$
$$Y^e \sim W_{1 \to Y}(X_1^e)^2 + W_{H \to Y} H^e + t_3(0, \sigma_y^2),$$
$$X_2^e \sim W_{Y \to 2} Y^e + W_{H \to 2} H^e + t_3(0, \sigma_2^2),$$

where $W_{H \to 1}, W_{1 \to Y}, W_{H \to Y}, W_{Y \to 2}, W_{H \to 2}, \sigma_y^2, \sigma_2^2$ are parameter vary across 4 experiment scenarios defined in Section 4.1. For each scenario, we generate $n_{\text{train}}^e = 1000$ training samples and $n_{\text{test}}^e = 500$ testing samples per environment. The results are presented in Fig 4

## B.2 DOMAINBED EXPERIMENT

### B.2.1 EXPERIMENTAL SETTINGS

We utilize the DomainBed dataset to evaluate the performance of invariance learning algorithms. DomainBed dataset (Gulrajani and Lopez-Paz, 2020) is a Pytorch testbed for domain generalization including multiple multi-domain datasets, baseline algorithms,

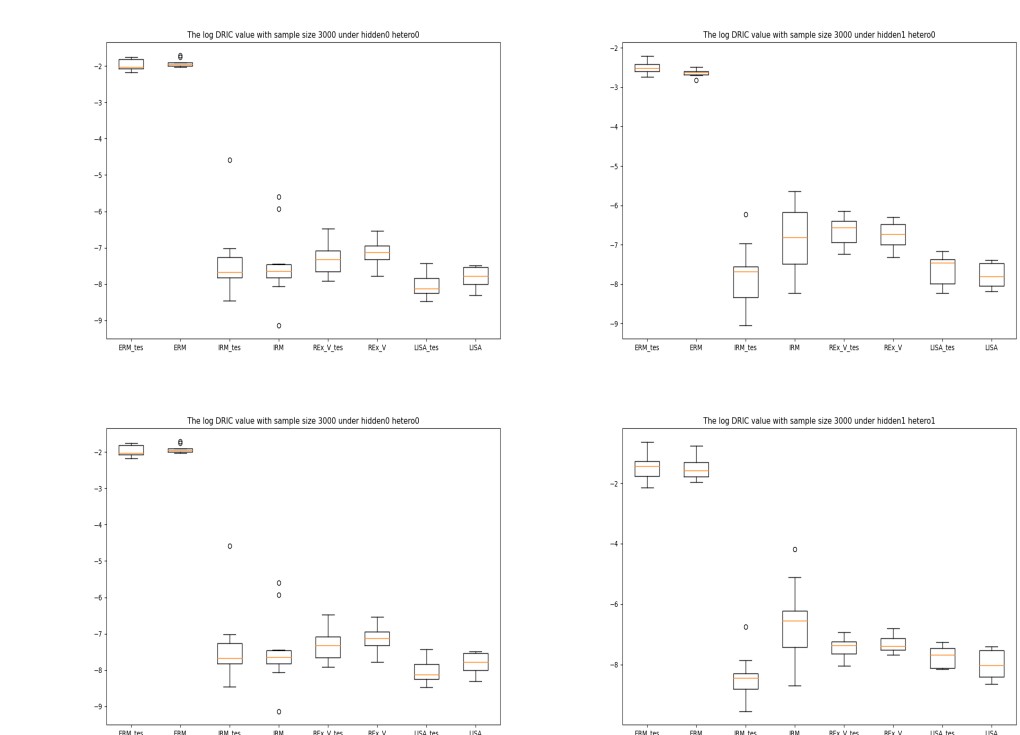

Figure 4: Nonlinear Synthetic Data

and model selection criteria. It streamlines reproducible and rigorous research in domain generalization. The detailed setup of the experiment is shown in Table 10, and the model architecture is shown in Table 11.

Table 10: Settings for DomainBed Experiment

| Condition | Parameter | Value |
|---|---|---|
| MNIST ConvNet | learning rate | 0.001 |
| | batch size | 64 |
| | generator learning rate | 0.001 |
| | discriminator learning rate | 0.001 |
| IRM | lambda | 10 |
| | iterations of penalty annealing | 500 |
| VREx | lambda | 10 |
| | iterations of penalty annealing | 500 |
| All | weight decay | 0 |
| | generator weight decay | 0 |

Table 11: MNIST ConvNet architecture

| # | Layer |
|---|-------|
| 1 | Conv2D (in=d, out=64) |
| 2 | ReLU |
| 3 | GroupNorm (groups=8) |
| 4 | Conv2D (in=64, out=128, stride=2) |
| 5 | ReLU |
| 6 | GroupNorm (8 groups) |
| 7 | Conv2D (in=128, out=128) |
| 8 | ReLU |
| 9 | GroupNorm (8 groups) |
| 10 | Conv2D (in=128, out=128) |
| 11 | ReLU |
| 12 | GroupNorm (8 groups) |
| 13 | Global average-pooling |

### B.2.2 CMNIST

We conduct our experiment on the CMNIST dataset, a variant of the MNIST handwritten digit classification dataset. In CMNIST, images are assigned binary labels (digits $< 5$ vs. digits $\geq 5$), and three environments are generated with varying strengths of spurious correlations between color and label ($+90\%, +80\%, -90\%$).

We evaluate five domain generalization algorithms—ERM, IRM, VREx, GroupDRO—using DomainBed's training framework. For each algorithm, we train five models (2 independent trials $\times$ 3 random hyper-parameter choices), resulting in a total of 60 models. During training, we use 2,333 examples for training and hold out 20% of data from each environment for validation. The best model for each algorithm is selected based on its validation performance in the training environment, and the validation accuracy are presented in Table 12.

| **Dataset**: ColoredMNIST | | | |
|---|---|---|---|
| Algorithm | +90% | +80% | -90% | Avg |
| ERM | $67.4 \pm 0.8$ | $68.6 \pm 0.5$ | $10.3 \pm 0.1$ | 48.7 |
| IRM | $70.7 \pm 1.3$ | $71.5 \pm 0.8$ | $9.9 \pm 0.1$ | 50.7 |
| GroupDRO | $69.3 \pm 0.0$ | $69.3 \pm 0.9$ | $9.3 \pm 0.3$ | 49.3 |
| VREx | $71.0 \pm 1.6$ | $70.3 \pm 0.5$ | $12.1 \pm 0.8$ | 51.1 |

Table 12: Validation Accuracy on CMNIST. $e = [+90\%, +80\%, -90\%]$ are environment labels. Avg denotes the average accuracy on all environments.

We then evaluate the selected models on a new split, consisting of 2,333 unseen examples from each environment. We calculate their test accuracy and DRIC values, as shown in Table 13 and Fig 3a. Our results show that all domain generalization algorithms achieve lower DRIC values than the baseline (DRIC for ERM = 1) and higher test accuracy, indicating that the more a model learns invariant relationships, the better it generalizes to unseen data.

| Algorithm | DRIC | Test Accuracy |
|---|---|---|
| ERM | 1.0000 | 48.12 |
| IRM | 0.2204 | 48.52 |
| GroupDRO | 0.2962 | 48.22 |
| VREx | 0.2660 | 48.52 |
| IB-IRM | 0.1900 | 50.60 |

Table 13: DRIC value and test accuracy of algorithms on CMNIST

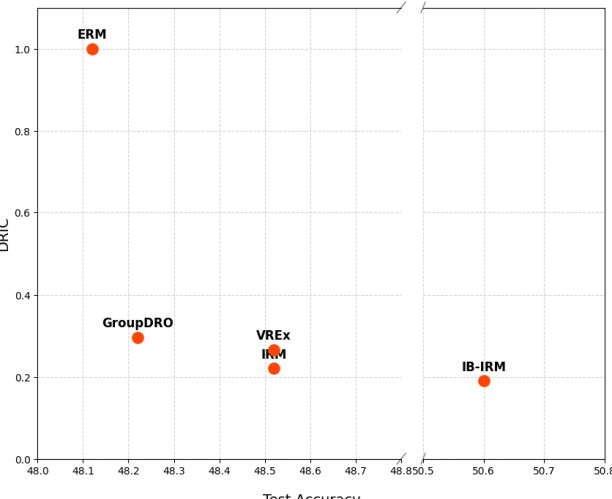

Figure 5: DRIC vs. Test accuracy

We further apply SVD method to reduce the high dimensionality of the input images to facilitate the calculation of density ratio, and obtain dimension of 128 for each flattened image in CMNIST dataset after SVD. In Table 14, we present the accuracy, DRIC value of three algorithms on 2 OOD-datasets, CMNIST. Table 14 shows that VREx DRIC value is smallest among three approaches, which shows that the VREx obtains the most invariant result among the three. This conclusion is consistent with the fact that VREx has the strongest invariance assumption.

Table 14: Reduce Dimension CMNIST

|          | Train Accuracy | Test Accuracy | DRIC  |
|----------|----------------|---------------|-------|
| **ERM**  | 0.7614         | 0.6732        | 0     |
| **IRM**  | 0.7372         | 0.7291        | -0.51 |
| **VREx** | 0.7451         | 0.7291        | -0.82 |

### B.2.3   PACS

We conduct experiments on the PACS dataset to evaluate invariance using the Expected Calibration Error (ECE). PACS is an image dataset with four different styles: photo, art painting, cartoon, and sketch. Each environment contains seven categories.

Following the experimental settings in Yoshida and Naganuma (2024), we set the training environments as cartoon, photo, and sketch, and use art as the testing environment. We record the ECE Variance values and test domain accuracy, with the results presented in Table 15. Our observations show that achieving lower ECE Variance is consistent with better prediction accuracy on unseen data. This suggests that learning invariant features and maintaining consistent calibration across environments enhances the model's generalization under environmental variations.

### B.3   REAL DATA EXPERIMENT

We listed the details of dataset and training setting for Real Data Section as below:

| Algorithm | Test Accuracy (%) | ECE Variance |
|---|---|---|
| ERM | 70.99 | 19.54 |
| IRM | 61.00 | 21.87 |
| GroupDRO | 69.67 | 17.95 |
| V-REx | 70.78 | 17.91 |

Table 15: ECE variance and test accuracy of algorithms on PACS

**Finance Data** [2] consists of factors in the U.S. stock market over five years. The data cleaning resulted in 37 features for company information and a target variable indicating stock price variation. The training data includes stock data from 2014 to 2016, while the testing data includes stock data from 2017 to 2018. Each year is an environment, with varying sample sizes per year.

**Law School Data** [3] includes student information for law school admission, with undergraduate GPA as the continuous target variable. The data contains 18 attributes after one-hot encoding. The training and testing sets include 1092 and 728 samples, respectively. For this dataset, environments were constructed based on the 'gender' attribute to simulate domain shifts. Following the protocol of prior work Zhao et al. (2022), we partitioned the data to create distinct training environments, each engineered to have a different correlation strength between gender and the target GPA.

**Adults Data** [4] is derived from the 1994 U.S. Census database to predict whether income exceeds 50K per year. It contains 14 features of personal information, which become 106 attributes after one-hot encoding, and a binary target. The training and testing sample sizes are 6,000 and 4,000 respectively. Similar to the Law School dataset, we use gender to define the environments.

**Bike Sharing Data** [5] contains the hourly and daily count of rental bikes, including 13 features and the target variable of total rental bikes count. Each season is treated as an environment, and data from 2011 and 2012 were used as training and test sets respectively.

For real data experiment, IRM, VREx, and LISA algorithms are employed training on Adults, Law school, Stock market, and Bike dataset. The experimental settings are listed in Table 16, and the detailed numerical results are listed in Table 17.

Table 16: Settings for House Price and Stock Market Experiment

| Parameter | Value |
|---|---|
| **IRM penalty** | 1e4 |
| **VREx penalty** | 1e4 |
| **L2 regularizer** | 0.001 |
| **Iterations of penalty annealing** | 1 |
| **Optimizer** | Adam |
| **Learning rate** | 1e-3 |

---

[2]https://www.kaggle.com/datasets/cnic92/200-financial-indicators-of-us-stocks-20142018
[3]https://github.com/algowatchpenn/GerryFair/blob/
[4]https://archive.ics.uci.edu/dataset
[5]https://www.kaggle.com/datasets/lakshmi25npathi/bike-sharing-dataset

Table 17: The $-\log_{10}$ DRIC value of different representation methods on training $(-\log_{10}(\hat{Q}_\phi))$ and testing $(-\log_{10}(\hat{Q}_\phi^t))$ for the Finance, Law School, Adults, and Bike Sharing datasets.

| Method | Dataset | $-\log_{10}(\hat{Q}_\phi)$ | $-\log_{10}(\hat{Q}_\phi^t)$ |
|---|---|---|---|
| IRM | Finance | 2.420 | 3.046 |
| | Law School | 1.992 | 2.796 |
| | Adults | 1.466 | 2.469 |
| | Bike Sharing | 1.272 | 1.307 |
| VREx | Finance | 3.000 | 3.155 |
| | Law School | 2.222 | 3.699 |
| | Adults | 2.456 | 4.019 |
| | Bike Sharing | 2.260 | 2.338 |
| LISA | Finance | 4.223 | 3.222 |
| | Law School | 2.268 | 3.699 |
| | Adults | 2.013 | 6.253 |
| | Bike Sharing | 5.804 | 4.000 |

### B.4 Additional Experiment: Comparison with invariant testing methods

In this section we compare our DRIC metric with group invariant techniques in Soleymani et al. (2025); Koning and Hemerik (2024); Chiu and Bloem-Reddy (2023), and domain classifier accuracy Ganin et al. (2016). We generate data according to a non-linear, non-Gaussian setting with environments $e = [0.1, 1, 10]$, each containing $n_{\text{train}}^e = 600$ training samples and $n_{\text{test}}^e = 400$ test samples. We train the algorithms ERM, IRM, VREx, MMD (Li et al., 2017), GroupDRO, and Mixup on $n_{\text{train}}^e$ using an MLP with hidden dimension 256 for 100 steps. After training, we evaluate each model on $n_{\text{test}}^e$, extract the final-layer representation $\hat{\phi}(x^e)$ for each algorithm, and concatenate it with the corresponding target $y^e$. To test invariance for each algorithm, we calculate DRIC value on test set, (i) perform statistical hypothesis testing on the joint distributions $(\hat{\phi}(x^e), y^e)$ to assess whether they are group invariant using techniques in Soleymani et al. (2025); Koning and Hemerik (2024); Chiu and Bloem-Reddy (2023), (ii) use domain classifier accuracy like Ganin et al. (2016) to predict environment label $e$ with augmented variable $(\hat{\phi}(x^e), y^e)$.

We begin with (i), where the symmetry group $G$ corresponds to the set of data domains. We randomly select one environment's data, denoted as $Z = [\hat{\phi}(x^e), y^e]$, and treat the other environments as its group-transformed counterparts, $G(Z) = [\hat{\phi}(x^{e'}), y^{e'}]$. To evaluate invariance, we first apply the Kernel Maximum Invariance Criterion (KMaxIC) proposed by Soleymani et al. (2025). As shown in the KMaxIC column of Table 18, we observe that ERM exhibits the the highest KMaxIC value of 0.0197, while VREx, MMD, and Mixup achieve significantly lower KMaxI values of 0.0048, 0.0071 and 0.0068 each, which indicates that these methods learn representations that are closer to being invariant.

Next, we divide environments into sub-environments and apply the subgroup-invariance test from Koning and Hemerik (2024). We use maximum mean discrepancy as the test statistic to evaluate whether the learned representations of each method remain distributionally invariant. We record the test statistic $T(GX)$ of each subgroup, and record the mean of $T(GX)$, denote as $T_{\text{sub}}$. A higher $T_{\text{sub}}$ indicates more sever violations of the invariance assumption, suggesting the method learns less stable representations. As shown in the $T_{\text{sub}}$ column of Table 18, ERM exhibits the highest $T_{\text{sub}}$ value (0.3857), indicating the weakest invariance. In contrast, VREx (0.1141), MMD (0.1248) and Mixup (0.1272) perform better, suggesting they learn more stable representations.

Lastly, we apply the non-parametric hypothesis test from Chiu and Bloem-Reddy (2023), using the Cramer–Wold (CW) projection method to assess distributional invariance. Under the null hypothesis $H_0$: the data distribution is invariant to the group action (data domains),

we compute the CW statistic and corresponding p-values. As shown in the CW, p-value, $H_0$ columns of Table 18, ERM obtains a p-value of 0, leading to rejection of $H_0$ and indicating non-invariance. In contrast, IRM, VREX, and Mixup obtain p-values of 0.16, 0.26, and 0.42 respectively, failing to reject $H_0$ and suggesting these methods learn invariant representations.

The total results are presented in Table 18, and the relations between DRIC and KMaxIC, $T_{\mathrm{sub}}$ are presented in Figure 6. These results consistently indicate that ERM method violate invariance assumption, while others are closer to invariance. We also observe a consistent monotonic trend relationship between the DRIC value and these invariant measures, indicating the alignment between the measures, and the reliability of DRIC in quantifying invariance.

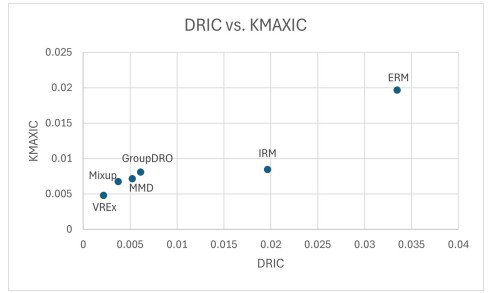
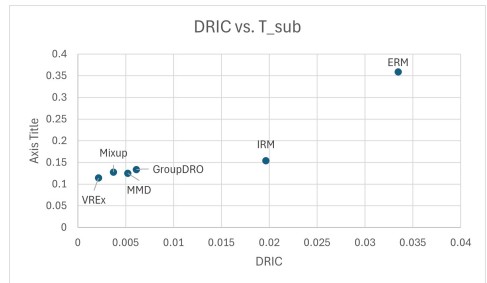

(a) DRIC vs. KMAXIC

(b) DRIC vs. $T_{\mathrm{sub}}$

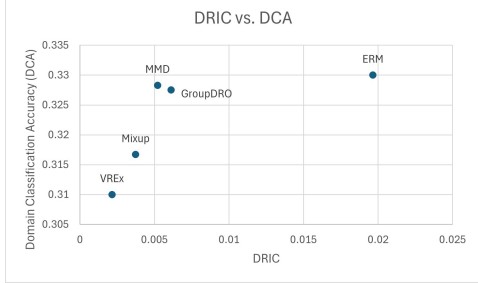

(c) DRIC vs. Domian Classification Accuracy (DCA)

Figure 6: DRIC values vs other metrics.

| Method | DRIC | DCA | KMAXIC | $T_{\mathbf{sub}}$ | CW | p-value | $H_0$: is invariant |
|---|---|---|---|---|---|---|---|
| ERM | 0.0335 | 1.0000 | 0.0197 | 0.3587 | 0.0003 | 0.00 | reject |
| IRM | 0.0197 | 0.3300 | 0.0084 | 0.1542 | 0.0000 | 0.16 | fail to reject |
| VREx | 0.0022 | 0.3100 | 0.0048 | 0.1141 | 0.0000 | 0.26 | fail to reject |
| MMD | 0.0052 | 0.3283 | 0.0071 | 0.1248 | 0.0000 | 0.50 | fail to reject |
| GroupDRO | 0.0061 | 0.3275 | 0.0081 | 0.1333 | 0.0000 | 0.20 | fail to reject |
| Mixup | 0.0037 | 0.3167 | 0.0068 | 0.1272 | 0.0000 | 0.42 | fail to reject |

Table 18: Results of DRIC vs Group Invariant Testing and Domain Classifier.

For (ii), we train an MLP classifier to predict the environment label $e$ using the augmented input $(\hat{\phi}(x), y)$. Since the ERM algorithm does not learn an invariant representation, we say $(\widehat{\phi}_{\mathrm{ERM}}(x), y)$ is environment dependent, and thus use $(\widehat{\phi}_{\mathrm{ERM}}(x), y)$ for training. The trained MLP classifier is then used to perform domain classification using the augmented variables obtained from each algorithm. The resulting classification accuracies and their relationship to DRIC are presented in column ACCUR in Table 18 and Figure 6. We obeserve that higher DRIC value corresponds to higher Domain Classification Accuracy (DCA) on $e$, indicating more environment information on $(\hat{\phi}(x), y)$, and the less invariance the method is.

### B.5 Additional Experiment: Robustness of DRIC

Theoretically, we have $0 \leq Q(\phi) \leq 1$. In addition, by Theorem 3.4, $Cov^2(Y, E) \leq Q(\phi)$ when the prediction risk is minimized. (i) Worst case: when $\phi(X) = X$, $Q(\phi) = 1$. Consider DRIC as defined in Equation (5), if $\phi(X) = X$, which means that the representation learns the original features, then it can not capture any domain invariant factors and thus the output of the algorithm is equivalent to the ERM. Therefore, $q_\phi(e, e') = q_\Upsilon(e, e')$ and $q_\phi(e, e) = q_\Upsilon(e, e)$, yielding $Q_\phi = 1$. (ii) Ideal case: We derive the lower bound of DRIC such that $Q(\phi) = Cov^2(Y, E)$ under the condition that $\mathbb{E}[Var(Y|\phi(X))] = 0$ almost surely. Therefore, the DRIC value is bounded in general cases, and will be upper bounded by 1 even in the worst case. Thus we can conclude that DRIC is a robust measurement.

Moreover, to validate the robustness of DRIC empirically, we conduct a simulation study under the nonlinear, non-Gaussian setting. We construct a family of representations of the form: $\hat{\phi}(X) = (X_1, \beta X_2)$, where $\beta$ varies from 0 to 1, $X_1$ is environment-invariant feature, and $X_2$ is environment-dependent variable. By varying $\beta$, we interpolate between different level of invariance in representation $\phi(x)$. From Figure 7 in the supplementary material, we observe that DRIC values increase monotonically from 0 to 1 as $\beta$ increases, showing that that DRIC is adaptive to the degree of invariance, yet remains bounded and stable even in extreme cases.

Overall, we can conclude that the DRIC metric is a robust measurement of invariance with both empirical and theoretical guarantee.

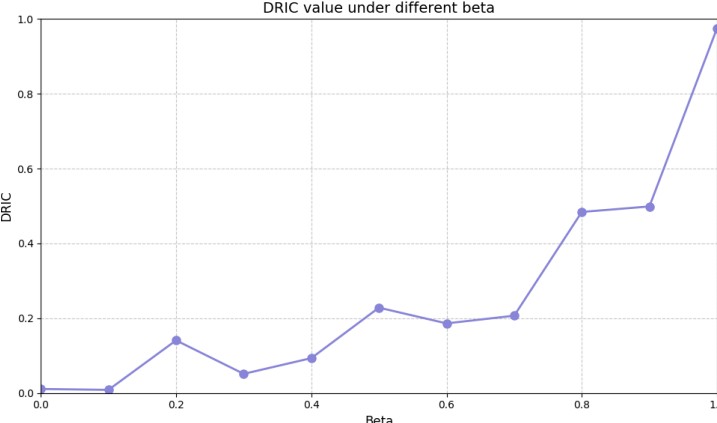

Figure 7: Robustness of DRIC.

### B.6 Additional Experiment: Comparison between GroupDRO and MixUP in linear model

To compare the performance of GroupDRO and MixUP algorithms, we fit a linear models by each, and compare DRIC value, loss and evaluate the $R^2$ value of the algorithms. We generate the data under the linear setting:

$$H^e \sim \mathcal{N}(0, e^2),$$
$$X_1^e \sim W_{H \rightarrow 1} H^e + \mathcal{N}(0, e^2),$$
$$Y^e \sim X^e + W_{H \rightarrow Y} H^e + \mathcal{N}(0, 1),$$
$$X_2^e \sim W_{Y \rightarrow 2} Y^e + W_{H \rightarrow 2} H^e + \mathcal{N}(0, e^2).$$

We train GroupDRO and MixUP model on this linear synthetic data and the results are shown in Table 19. In Table 19, we show DRIC value and test accuracy for Mixup and GroupDRO methods under linear setting. The Tabel 19 shows that the Mixup and GroupDRO methods produce similar test loss while DRIC value are 0.5113, 0.6875 respectively. The corresponding $R^2$ value for the two methods are 0.6470, 0.6325 respectively, which demonstrate that the better performance of DRIC relies on higher explained variance.

Table 19: Comparison of Mixup and GroupDRO

| Metric | Mixup | GroupDRO |
|--------|-------|----------|
| Q Test | 0.5113 | 0.6875 |
| Test $R^2$ | 0.6470 | 0.6325 |
| Test Loss | 0.3530 | 0.3675 |

## B.7 Additional Experiment: DRIC's Correlation with OOD Risk

To verify that DRIC tracks representation-level invariance and to show the relationship between test risk and DRIC, we swept the VREx penalty $\lambda$ on the SEM synthetic data generated in Section 4.1 and recorded train/test MSE and unnormalized DRIC.

Table 20: VREx penalty ($\lambda$) sweep results on the SEM synthetic data.

| Penalty ($\lambda$) | Train MSE | Test MSE | DRIC (unnormalized) |
|---------------------|-----------|----------|---------------------|
| 10 | 0.20538 | 3.19179 | 1.297e-5 |
| 100 | 0.21147 | 3.16218 | 1.0355e-4 |
| 1000 | 0.21586 | 3.11066 | 7.786e-5 |
| 10000 | 0.25938 | 2.61928 | 5.913e-5 |
| 100000 | 0.63894 | 1.45442 | 3.385e-5 |
| 1000000 | 0.70784 | 1.35925 | 2.94e-6 |

**Observation.** We can observe that test-MSE drops from 3.19 to 1.36 as DRIC falls, despite the training-MSE rising. Thus, the model that looks worst on training loss is actually the most robust once DRIC is considered. This sweep shows that DRIC and OOD accuracy move together, but non-linearly.

Plotting (Accuracy, DRIC) against $\lambda$ shows:

- Up to the knee at $\lambda \approx 10^5$, increasing $\lambda$ lowers both DRIC and OOD test error.
- Beyond that knee, accuracy plateaus while DRIC continues to shrink.

A reasonable stopping rule is therefore: *increase $\lambda$ until accuracy saturates or a target DRIC is reached.*