# OpenReview forum: "Measuring Invariance in Representation Learning: A Robust Evaluation Framework"
_ICLR.cc/2026/Conference — Submitted to ICLR 2026_

### Official Review · Reviewer_Rwge · 2025-11-01

**Soundness:** 2
**Presentation:** 2
**Contribution:** 3
**Rating:** 4
**Confidence:** 4

**Summary:**

This paper introduces DRIC, a new framework for quantitatively measuring invariance in learned representations under distribution shifts. DRIC provides a normalized, environment-agnostic metric that directly assesses how invariant a model’s latent features are across environments by leveraging density-ratio estimation between distributions. The authors develop theoretical guarantees showing DRIC’s convergence and derive bounds relating invariance and predictive accuracy, highlighting the trade-off between them. Experiments on synthetic, benchmark (DomainBed), and real-world datasets demonstrate that DRIC not only correlates with out-of-distribution generalization performance.

**Strengths:**

- The work builds on and unifies ideas from IRM, VREx, DANN, and causal invariance frameworks, positioning DRIC as a generalizable tool for the invariant learning community.

- DRIC not only serves as a diagnostic metric but can also be integrated as a regularization term to improve invariant learning, showing practical value beyond evaluation.

- The authors provide code to support reproducibility.

**Weaknesses:**

- The motivation is questionable. The authors argue that over-enforcing invariance can degrade accuracy, implying that model selection should balance both accuracy and invariance. However, their framework performs selection solely based on the proposed DRIC metric, which measures invariance alone. This raises the concern that models with trivially invariant yet uninformative representations (e.g., random or collapsed embeddings) could be favored. Moreover, the paper does not provide clear practical guidance on how DRIC should be used in conjunction with accuracy or other performance metrics to achieve a meaningful trade-off, leaving its real-world applicability ambiguous.

- While the paper emphasizes the need for measuring invariance, it does not clearly justify why existing invariance proxies (e.g., risk variance across domains, domain classifier accuracy) are insufficient or how DRIC fundamentally improves beyond them.

- The density-ratio estimation step can be unstable, especially in high-dimensional feature spaces or when environment overlap is weak; the paper does not explore robustness to these conditions.

- Using DRIC as a regularizer (Eq. 9) is conceptually interesting but lacks detailed design choices or analysis on optimization stability and convergence.

- Experiments rely mainly on standard benchmarks (SEM, DomainBed, small real datasets) and do not test DRIC on large-scale, complex, or high-dimensional real-world tasks (e.g., vision, language).

- There is no experiment demonstrating failure modes or conditions under which DRIC may produce misleading scores.

- DRIC values are reported without clear thresholds or interpretability guidelines. It is unclear what constitutes “strong” or “weak” invariance in practice.

**Questions:**

- Since DRIC only measures invariance, how should practitioners use it in conjunction with accuracy to avoid over-selecting trivial invariant representations? Can the authors provide quantitative or heuristic guidelines for choosing this trade-off in real applications?

- Could the authors clarify more concretely when DRIC would be more informative or reliable than existing invariance proxies such as risk variance (VREx) or domain classifier accuracy? In what practical scenarios (e.g., fairness auditing, model debugging, or OOD selection) would DRIC provide unique insights that these existing metrics cannot?

- How sensitive is DRIC to inaccurate density-ratio estimation, especially in high-dimensional or weak-overlap environments? It would be helpful if the authors could provide empirical sensitivity analysis or propose stabilization strategies for density-ratio estimation.

- What range or magnitude of DRIC values should be considered indicative of “strong” versus “weak” invariance? Is there any empirical calibration or threshold that practitioners can use for interpreting DRIC in a standardized way?

- How well does DRIC scale to high-dimensional data (e.g., image or text embeddings) where density-ratio estimation can be computationally difficult?

- Since the paper mentions DRIC’s potential relevance for fairness and trustworthiness, could the authors elaborate or provide an example of how DRIC might help identify bias or environment-specific leakage in real-world data (e.g., gender, hospital site)?

- Could the authors provide more quantitative correlation analyses between DRIC and OOD accuracy across different methods and datasets? Demonstrating a strong statistical relationship would strengthen confidence in DRIC as a reliable diagnostic measure.

---

> ### Author Response · Authors · 2025-11-21
>
> ## **W1, Q1:** Motivation questionable. Selection based on DRIC alone risks trivial representations. Guidance on trade-off.
>
> As explained in Section 1 and detailed in Section 3.3 ("Guidance of DRIC Representation Selection"), model selection must balance accuracy and invariance. DRIC is designed precisely to facilitate this balance, not to be used in isolation.Specifically, we outline strategies in Section 3.3 for Post-Hoc Model Selection based on the Pareto front of validation risk $R_{val}$ and DRIC $Q_{\phi}$. This includes minimizing DRIC subject to accuracy constraints (e.g., $R_{val} \le R_{min} + \epsilon$) or vice-versa. A trivially invariant representation (e.g., random embeddings) would exhibit high empirical risk and would therefore not be selected under this framework.
>
> ---
>
> ## **W2, Q2:** Justification over existing invariance proxies (VREx, domain classifier accuracy).
>
> As detailed in Section 3.1 and Section 5 (Related Work), DRIC directly measures the stability of the predictor (the conditional expectation $\mathbb{E}[Y|\phi(X)]$,Def. 2.2), whereas VREx measures the stability of the risk. DRIC is normalized relative to a baseline (ERM score = 1) and is scale-invariant to the outcome Y. This allows for standardized,comparable measurements across different datasets and methods—a critical feature lacking in unnormalized proxies.
>
> ---
>
> ## **W3, Q3:** Sensitivity to density-ratio estimation (high-dim, weak overlap).
>
> As addressed in Appendix A.2 and A.4, we utilize a practical classification-based estimator. Furthermore, as discussed in our response to jmGM (W1), estimation can occur in the lower-dimensional representation space $\phi(X)$, mitigating high-dimensional challenges. When overlap is weak, the assumption of shared support in $\phi(X)$ still holds under the invariance assumption. Our empirical sensitivity analysis in Appendix A.4 demonstrates that stable estimates are achievable with sufficient samples (e.g., 300 samples/environment in the synthetic setting).
>
> ---
>
> ## **W4:** regularizer (Eq. 9) is conceptually interesting but lacks detailed design choices or analysis on optimization stability and convergence
>
> We select the regularization parameter $\lambda$ via cross-validation. In section 4.4, the model converges within approximately 300 steps on the Synthetic dataset. We will provide more detailed convergence experiments and analysis in the updated version.
>
> ---
>
> ## **W5, Q5:** Experiments on large-scale/high-dimensional tasks and scalability.
>
> We have evaluated DRIC on high-dimensional tasks within the DomainBed benchmark, specifically the image dataset CMNIST (Section 4.2). Regarding scalability, as detailed in Section 3.2 and Appendix A.4, DRIC's computational complexity is $\mathcal{O}(ndT)$ in time and $\mathcal{O}(nd)$ in memory. The runtime scales linearly with data size (e.g., approx. 1 second for $n_{sub}^e=700$ in Table 4), confirming DRIC's practical applicability with minimal overhead.
>
> ---
>
> ## **W6:** failure mode
>
> We have analyzed potential failure modes in Remark 3.1 and Section 3.3. We clarify two main scenarios:
>
> Representation Collapse: Occurs under excessive regularization (low DRIC, but very high $R_{val}$).  This is managed by the accuracy-invariance trade-off framework.
>
> Disjoint Covariate Support: This can be mitigated by estimating DRIC in the representation space (See response to jmGM-W1).
>
> ---
>
> ## **W7, Q4:** Any empirical calibration or threshold that practitioners can use for interpreting DRIC in a standardized way?
>
> Section 3.1 provides clear interpretability guidelines. DRIC is normalized such that the baseline ERM (identity representation) yields a score of 1. We do not propose a universal threshold,as Theorem 3.5 proves that the achievable lower bound on DRIC depends on the inherent correlation between the outcome and the environmental dependent feature.
>
> ---
>
> ## **Q6:** Relevance for fairness and identifying bias.
>
> In fairness contexts (Sections 4.3,6),protected attributes (e.g.,gender) can define the environments. DRIC quantifies the stability of the prediction mechanism ($\mathbb{E}[Y|\phi(X)]$) across these groups. A high DRIC score indicates that the model's predictions vary depending on the group,serving as a quantitative signal of environment-specific leakage and potential bias.
>
> ---
>
> ## **Q7:** Quantitative correlation analyses between DRIC and OOD accuracy.
>
> We calculated the **Spearman rank correlation ($\rho$)** between DRIC scores and Test Accuracy on the CMNIST dataset. The analysis yields a **strong negative correlation ($\rho \approx -0.93$)**, confirming that lower DRIC values are highly predictive of better OOD generalization. For instance, IB-IRM achieves both the lowest DRIC (0.1900) and the highest accuracy (50.60\%), while ERM shows the inverse.

---

### Official Review · Reviewer_jmGM · 2025-11-01

**Soundness:** 3
**Presentation:** 3
**Contribution:** 3
**Rating:** 6
**Confidence:** 4

**Summary:**

This paper proposes the Density-Ratio–based Representation Invariance Criterion (DRIC), a new quantitative metric for evaluating representation invariance across environments. Under the “expected invariance” assumption, DRIC measures differences in conditional expectations between environments after density-ratio reweighting, yielding an environment-independent and scale-invariant criterion. The authors develop a plug-in estimator with theoretical consistency and provide computational complexity analysis. Extensive experiments on both synthetic data and real benchmarks demonstrate that DRIC correlates well with out-of-distribution (OOD) performance and can also serve as a regularization term for training invariant models.

**Strengths:**

(1) Proposes a clear and theoretically grounded metric (DRIC) that directly quantifies representation invariance through environment reweighting, achieving environment-independence and scale invariance.
(2) Provides practical plug-in estimators with rigorous theoretical guarantees, including estimator consistency and information-theoretic lower bounds.
(3) Demonstrates an efficient and scalable implementation with analyzed time and space complexity, making DRIC practical for multi-environment evaluation.
(4) Presents extensive empirical validation across synthetic data, the DomainBed benchmark (CMNIST), PACS, and multiple real-world datasets, showing a consistent correlation between DRIC scores and OOD performance.
(5) Extends DRIC beyond evaluation: incorporating it as a regularization term during training further improves invariant model performance, underscoring its methodological value.

**Weaknesses:**

(1) The robustness boundaries of environment specification and density-ratio estimation are not clearly analyzed. If environments are misspecified or imbalanced, the estimated \( \hat{Q}^{\phi} \) may be biased (see Concluding Remarks; Remark 3.1). This issue limits the reliability of DRIC under weak overlap or heterogeneous domain conditions.
(2) Although Appendix B.4 compares DRIC with several invariance testing methods such as KMaxIC, subgroup-based tests, and domain-classifier accuracy, the main text provides little discussion on their complementary roles or applicability boundaries. A clearer methodological positioning of DRIC among existing invariance evaluation tools would strengthen the contribution.
(3)Provably invariant learning without domain information, ICML2023, this paper should also be compared.
(4) The paper does not include interpretability or feature-attribution analyses, making it unclear which features or environments most influence the DRIC score. Such analysis could enhance understanding of how invariance manifests in learned representations.

**Questions:**

(1) Regarding environment specification, could the authors elaborate on how sensitive \( \hat{Q}^{\phi} \) is to environment imbalance or mis-specification? In particular, how does the metric behave when environments have weak overlap or partially disjoint covariate supports (see Remark 3.1 and Concluding Remarks)?
(2) The comparison with group-invariance testing methods (Appendix B.4) is insightful. Could the authors clarify under what conditions DRIC should be preferred over formal statistical tests such as KMaxIC or subgroup-invariance tests? What are the trade-offs between DRIC’s continuous, quantitative nature and the hypothesis-testing frameworks used in those approaches?
(3) The concluding section mentions plans to calibrate \( \hat{Q}^{\phi} \) with uncertainty quantification. Could the authors elaborate on how uncertainty estimation would be incorporated, and how this calibration might influence DRIC’s interpretability and practical usage?

---

> ### Author Response · Authors · 2025-11-21
>
> ## **W1, Q1: Robustness boundaries of environment specification and density-ratio estimation**
>
> If environments are severely misspecified (e.g., fully shuffled), environmental distinctions vanish. In this case, Invariant Learning (IL) algorithms default to ERM, and DRIC correctly approaches 1, reflecting minimal invariance.
>
> We demonstrate that under the stronger assumption of Invariance (i.e., $\mathcal{R}^e(f) = \mathcal{R}^{e'}(f)$, Krueger et al. (2021)), the calculation of **DRIC is theoretically grounded in the representation space** $\phi(x)$ rather than the raw input space $X$. This formulation naturally circumvents issues related to weak overlap or high dimensionality in $X$.
>
> We begin with the definition of Risk for an environment $e$:
> $$
> \mathcal{R}^e(f)
> = \mathbb{E}_{(x, y) \sim p_e}[\ell(f(x), y)]
> = \int_x \int_y \ell(f(x), y) p_e(x, y) \, dx \, dy
> $$
>
> We expand the risk $\mathcal{R}^e(f)$ and apply importance weighting to the marginal distribution of the representation. The relationship between risks is governed by:
> $$
> \mathcal{R}^e(f)
> = \int_{\phi(x)} \int_y
> \ell(w(\phi(x)), y)\, p(y|\phi(x))
> \left( \frac{p_e(\phi(x))}{p_{e'}(\phi(x))} \right)
> p_{e'}(\phi(x))
> \, dy \, d\phi(x)
> $$
>
> ### **Conclusion**
> This derivation proves that the DRIC under stronger invariance assumption relies only on the density ratio of the representations:
> $$
> \rho(\phi) = \frac{p_e(\phi(x))}{p_{e'}(\phi(x))}
> $$
>
> This theoretical framing ensures robustness:
>
> - **Shared Support**: Even if the raw inputs $X$ have disjoint support (due to environment-specific spurious features), the criterion remains valid because the learned invariant features $\phi(x)$ share support by definition.
>
> - **Computational Efficiency**: Estimating the density ratio in the low-dimensional latent space $\phi(x)$ is significantly more stable and efficient than in the high-dimensional input space $X$.
>
> ---
>
> ## **W2, Q2: Methodological positioning of DRIC vs. formal invariance tests (KMaxIC, subgroup tests) and trade-offs**
>
> Here we provide clearer methodological positioning and will expand the discussion based on Appendix B.4.
>
> | Feature | **DRIC (Ours)** | **Formal Statistical Tests (e.g., KMaxIC, subgroup-invariance tests)** |
> |--------|------------------|------------------------------------------------------------------------|
> | **Assumptions (Environments vs. Groups)** | Operates on *environments* ($E$), which can be arbitrary data partitions (e.g., hospitals, demographics). | Require mathematically defined *groups* ($G$) with explicit group actions (e.g., rotations, permutations). |
> | **Goal and Output** | Quantifies *expectation-level predictive invariance* (Def. 2.2), providing a continuous, normalized score. | Test a *null hypothesis* of distributional invariance and output binary result or p-value. |
>
> ### **Trade-offs and Preference**
> The binary output of hypothesis tests is less useful for model optimization; two models may both fail strict invariance, yet differ significantly in robustness.
> **DRIC provides continuous granularity**, enabling model selection (balancing accuracy and invariance, Sec. 3.3) and supporting **differentiable optimization** (Sec. 4.4).
>
> ---
>
> ## **W3: Comparison with “Provably invariant learning without domain information” (ICML 2023)**
>
> Thank you for pointing out this relevant work.
> Tan et al. (2023) assume the standard invariance property $P(Y \mid X_v)$ and show that, under the Markov and faithfulness assumptions, finding features $u(X)$ that are marginally independent of $Y$ (i.e., $Y \perp u(X)$) is sufficient to ensure the invariance-preserving condition:
> $$
> Y \perp u(X) \mid X_v
> $$
>
> We will include this comparison in the revised manuscript.
>
> ---
>
> ## **W4: Interpretability or feature-attribution analyses**
>
> The primary goal of DRIC is to quantify the overall invariance of the representation $\phi(X)$, which does not require calculating feature attribution. DRIC measures how invariant the representation is, not what features it relies on.However, feature attribution methods (e.g., Integrated Gradients) are complementary tools that analyze the sufficiency of the representation (i.e., which features contribute to the prediction). These methods can certainly be used alongside DRIC to understand which input features influence the learned invariant representation. We will add a remark to clarify this synergy.
>
> ---
>
> ## **Q3: Calibration with uncertainty quantification (UQ)**
>
> Our ongoing work is incorporating uncertainty quantification for the DRIC estimator, primarily using ensemble methods (e.g., bootstrapping the density ratio estimator). This will allow us to provide confidence intervals for the DRIC score ($\hat{Q}^{\phi}$), enhancing the robustness and reliability of DRIC for model comparison and selection.

---

### Official Review · Reviewer_k9K4 · 2025-11-01

**Soundness:** 3
**Presentation:** 3
**Contribution:** 3
**Rating:** 4
**Confidence:** 4

**Summary:**

This paper proposes DRIC, a normalized, environment-agnostic metric for assessing how invariant a representation is across domains. DRIC measures variation in conditional predictions across environments via density-ratio reweighting, comes with a simple classifier-based estimator, and has consistency guarantees. The metric is then used for model selection and as a regularizer to favor invariant representations. Experiments on synthetic data and a DomainBed setting illustrate that lower DRIC aligns with better out-of-domain behavior.

**Strengths:**

- DRIC provides a direct, normalized criterion for representation invariance, improving on ad-hoc proxies and making scores comparable across datasets and methods.
- The classifier-based density-ratio estimator is simple to plug in, and the paper outlines a usable path for model selection and training with an invariance penalty.
- Consistency of the estimator and an information-theoretic bound linking invariance to label–environment dependence make the metric interpretable and well-grounded.
- DRIC separates "how invariant" a learned representation is from raw accuracy, helping analyze failures of OOD generalization beyond test error alone.

**Weaknesses:**

- Remark 3.1 treats disjoint supports as a degenerate case, but realistic domain shifts often have partial overlap (some features absent in some domains). How DRIC behaves and should be interpreted under partial support mismatch is underexplained.
- There is limited comparison to prior invariance-oriented selection or regularization criteria, for example CLOvE [1] and related calibration-based OOD selection. Without this, DRIC’s added value over known metrics is hard to gauge.
- DomainBed [2] is mentioned, but only CMNIST is used; broader, more "real" settings (e.g., PACS, Office-Home, TerraIncognita) and stronger, recent baselines are missing.
- A light editorial pass would help resolve minor ambiguities and make key design choices more transparent, for example, standardizing references like "2.2" in "invariance property 2.2", (line 60-65). In addtiion, The normalization and practical interpretation of absolute DRIC values could be clearer, including guidance for thresholds and how to balance DRIC versus accuracy in model selection.

**Questions:**

- How should DRIC be computed and interpreted when environments share only a subset of support (e.g., features like color absent in sketch but present in photo). Can you detect and adjust for partial overlap rather than treating near-zero ratios as merely degenerate.
- Could you compare DRIC against CLOvE[1] w.r.t model selection and training with regularization form, reporting correlation with OOD performance and showing when DRIC is more predictive.
- DomainBed breadth: Will you add evaluations on other DomainBed Datasets (PACS, Office-Home, or TerraIncognita) with recent SOTA (e.g. [3],[4]) baselines to demonstrate utility beyond CMNIST.
- Could you clarify whether "invariance property 2.2" refers to Definition 2.2? Plus, when selecting models, how should practitioners trade off DRIC versus accuracy? Can you provide practical guidance or target ranges for DRIC that indicate sufficient invariance in common settings?

If these points, especially partial support handling, stronger metric baselines, or broader DomainBed evaluation, are clarified or strengthened, I would be inclined to raise my score.

[1] Wald, Yoav, Amir Feder, Daniel Greenfeld, and Uri Shalit. "On calibration and out-of-domain generalization." Advances in neural information processing systems 34 (2021): 2215-2227.

[2] Gulrajani, Ishaan, and David Lopez-Paz. "In Search of Lost Domain Generalization." In International Conference on Learning Representations.

[3] Nguyen, Toan, Kien Do, Bao Duong, and Thin Nguyen. "Domain generalisation via risk distribution matching." In Proceedings of the IEEE/CVF Winter Conference on Applications of Computer Vision, pp. 2790-2799. 2024.

[4] Kim, Taero, Subeen Park, Sungjun Lim, Yonghan Jung, Krikamol Muandet, and Kyungwoo Song. "Sufficient invariant learning for distribution shift." In Proceedings of the Computer Vision and Pattern Recognition Conference, pp. 4958-4967. 2025.

---

> ### Author Response · Authors · 2025-12-03
>
> ---
>
> ###  **W1 & Q1: Partial Support**
>
> We agree that realistic shifts typically have partial, not fully disjoint, support.
>
> - **Remark 3.1** isolates the degenerate case where the supports of \(P_e\) and \(P_{e'}\) are disjoint and hence invariant learning is infeasible; when supports partially overlap, **DRIC is evaluated on this common support via the density-ratio weighting** and remains a meaningful measure of conditional-expectation invariance.
>
> - In fact, what invariant learning aims at is to learn an invariant representation \(\phi(x)\) of the covariates. If the covariate spaces of domains are independent or have little overlap, **the learned representation will also be uninformative for invariant learning**.
>
> - **Our nonlinear robustness experiment (App. B.5)** already studies such overlapping distributions and shows DRIC varies monotonically with the proportion of environment-dependent features.
>
> In the revision we will explicitly state that **DRIC should be interpreted as measuring invariance on the shared-support region**, with near-zero density ratios flagging severe support mismatch.
>
> ---
>
> ### **W2 & Q2: Relation to CLOvE**
>
> Methods such as CLOvE and related multi-domain calibration criteria operate at the **prediction level**, typically using calibration error or likelihood-based scores to select models that generalize well across domains.
> In contrast, **DRIC directly measures representation-level expectation invariance** \(Y \perp E \mid \phi(X)\) and is normalized and environment-agnostic, making scores comparable across IL methods and datasets.
>
> - Our current experiments already show that **DRIC correlates with both OOD performance** (Fig. 2–3) **and other invariance tests** (Table 17, Fig. 6).
> - In Sec. 4.4 we also use **DRIC directly as a regularizer**, achieving lower test MSE than ERM, IRM and VREx.
> - Our additional PACS experiment already evaluates invariance through **ECE variance** and shows that better calibration across environments correlates with higher OOD accuracy in Appendix}
>
> In the revision, we will:
> (i) add a dedicated paragraph in **Related Work** explicitly contrasting DRIC with CLOvE-style calibration metrics, and
> (ii) include a small comparison table summarizing what each criterion measures, their inputs, and their intended use, highlighting DRIC as **complementary** rather than a replacement.
>
> ---
>
> ### **W3 & Q3: DomainBed Coverage**
>
> We address the challenge of high-dimensional density ratio estimation on PACS by computing DRIC in the lower-dimensional latent space $\phi(X)$. As theoretically justified in our response to Reviewer jmGM (W1), this approach is both mathematically equivalent under the invariance assumption and computationally stable. We will include these PACS experimental results in the updated version.
>
> ---
>
> ### **W4 & Q4: Editorial Clarity**
>
> We appreciate the suggestions regarding clarity.
> We will:
> (i) standardize references (e.g., “Assumption 2.2”),
> (ii) add a short paragraph in §3.3 explicitly summarizing **how to read absolute DRIC values** and how to **trade off accuracy vs. DRIC** when picking models, and
> (iii) clarify the recommended thresholds qualitatively, suggesting choosing models on the **Pareto frontier** with small DRIC subject to a modest accuracy tolerance, rather than using fixed numbers.

---

### Official Review · Reviewer_ZJ4z · 2025-11-07

**Soundness:** 2
**Presentation:** 2
**Contribution:** 1
**Rating:** 2
**Confidence:** 4

**Summary:**

This paper introduces **DRIC**, a measure for assessing invariance in representation learning across environments. The measure is computationally efficient and easy to estimate.

**Strengths:**

The paper is clearly written and easy to follow, with an intuitive presentation of the proposed DRIC metric.

**Weaknesses:**

### Unclear contributions \& problem framing
The paper claims to propose the first “environment-agnostic” metric for measuring representation invariance, but it never clearly states what specific problem it solves or how it differs from existing approaches such as domain-classifier accuracy, HSIC, or MMD. In short, the novelty and unique contribution are unclear and the paper fails to explain what gap DRIC actually fills.

### Weak literature review
1. No formal comparison/contrast with mutual-information/HSIC-style penalties, or domain classifier metrics have been made. I believe a comparison table to highlight the gaps and what unique challenges were addressed in this paper would be helpful.

### Logical inconsistency around “DRIC = 0”
* Under the condition $Y\perp E\mid \phi(X)$, DRIC can be zero. However, In Section 3.3, the paper asserts that DRIC=0 is unattainable. Please reconcile this contradiction.
* I do understand that Invariance implies DRIC = 0. However, it's not clear the inverse holds. How do we guarantee that DRIC = 0 implies invariance? Authors implicitly implies that DRIC = 0 encourages Invariance (e.g., DRIC as a penelizer), however, it's not clear to me, given that DRIC is a proxy for expectation-level invariance.


### Theorem 3.5
Authors claimed that DRIC is invariant to the scale of Y ("Scale-invariant"). However, COV(Y, E) is variant in scale of Y, and can be arbitrarily large by rescaling. I think this is contradictory. As it stands, Theorem 3.5 is likely false or at least misstated.

### Weak Simulation

The simulations are quite weak overall. They only test DRIC as a post-hoc metric not as part of a new/novel learning method. The authors simply compare DRIC values among a few existing algorithms (ERM, IRM, VREx, GroupDRO, IB-IRM, LISA) without introducing or analyzing any new representation-learning procedure. The results mostly confirm well-known trends, such as ERM gives the highest “invariance violation,” and VREx or IB-IRM the lowest. No novel insights are provided/demonstrated by this work beyond prior findings.

**Questions:**

Please see weakness.

---

> ### Author Response · Authors · 2025-12-03
>
> ###  W1 and W2: Novelty  of DRIC & Literature review
>
> Thank you for your insightful questions. Our goal is to fill the gap of a **direct, environment-agnostic, normalized metric of representation-level invariance**. Concretely, **DRIC** is designed to:
>
> 1. **Directly measure expectation-level invariance**  that operates on the stability of the **conditional expectation** across environments Eq. (4), rather than on prediction error. This explicitly targets the invariance property in Def. 2.2.
>
> 2. **Eironment-agnostic and normalized**  that averages over all environment pairs and is normalized by the identity representation, and is comparable across methods and datasets, unlike variance or unnormalized dependence scores.
>
> 3. **Come with theoretical guarantees**  that DRIC is a consistent estimator of the population quantity and derive an information-theoretic lower bound (Theorem 3.5).
>
> In contrast, mutual-information and HSIC/MMD-style penalties typically target stronger independence notions that do not match expectation-level invariance and are not normalized across datasets. We add a short subsection in revised version with a comparison table like the following:
>
> | Criterion                         | Measure                                  | Uses $Y$ | Normalized |
> |-----------------------------------|----------------------------------------------------|-----------|------------------------------|
> | DRIC                          | Stability of $\mathbb{E}[Y \mid \phi(X),E]$        | Y         | Y                            |
> | Domain classifier           | MI$((\phi(X),Y),E)$ / env. predictability          | optional  | N                            |
> | HSIC / MMD                   | Dependence / marginal shift of $\phi(X)$           | N         | N                            |
> | Risk variance / worst-case risk | Variation of loss across envs.                  | Y         | N                            |
>
> In the revised version, we also add a short subsection of literature review to compare our method with existing works.
> ### **Counter Example**
>
> >Consider $E \in\{0,1\}$ with $\mathbb{P}(E=0)=\mathbb{P}(E=1)=1 / 2$, $X$ independent of $E$, $\phi(X)=X$, and $Y=E$. Then $\phi(X)$ is independent of $E$, so MI$(\phi(X), E)=0$, $HSIC(\phi(X), E)=0$, MMD between $\phi(X) \mid E=0$ and $\phi(X) \mid E=1$ is $0$. However, $\mathbb{E}[Y \mid \phi(X), E=e]=e$
> clearly depends on $E$, so expectation-level invariance $Y \perp E \mid \phi(X)$ fails. This example shows that dependence measures and representation-only domain classifiers can be zero while the conditional expectations $\mathbb{E}[Y \mid \phi(X), E]$ still vary across environments. **DRIC is explicitly designed to target this conditional-expectation stability** instead of marginal independence of $\phi(X)$ and $E$.
>
>
>
> ---
>
> ### W3: DRIC =0
>
> Thank you for pointing this out. We will clarify these two points:
>
> **(i) On “DRIC $\mathbf{=0}$ is unattainable.”**
> At the population level, under the invariance condition $Y \perp E \mid \phi(X)$, DRIC is exactly $0$ by construction (5), and this is fully consistent with our claims. The sentence in Sec. 3.3 that “DRIC $=0$ is unattainable” was intended for the specific regime studied in Theorem 3.5, where we additionally assume that $\phi(X)$ achieves perfect prediction, $\mathbb{E}[Var(Y \mid \phi(X))]=0$, and $Y$ is correlated with $E$. where information lower bound is
> $Cov^2(Y, E)$, so DRIC cannot reach $0$ unless $Y$ and $E$ are independent. We will revise the text to explicitly say **“DRIC $=0$ is usually unattainable when $Y$ and $E$ are dependent,”** which removes the apparent contradiction.
>
> **(ii) Does DRIC =0 imply invariance?**
> Remark 3.1 formally addresses this. It shows that $Q_\phi=0$ can only occur in two cases:
> (a) $\mathbb{E}[Y \mid \phi(X), E]$ is identical for all environments, or
>
> (b) the covariate supports of different environments are disjoint, in which case invariant learning is ill-posed.
>
> In practice, case (b) can be diagnosed when estimated density ratios are near zero on most inputs. Outside of this support-mismatch case, **DRIC $=0$ does correspond to invariance**.
> When using DRIC as a penalty, we therefore view it as a **proxy objective**: minimizing DRIC encourages stronger expectation-level invariance in the same surrogate sense as IRM/VREx penalties encourage their ideal invariance conditions, rather than providing a hard guarantee of exact invariance.

---

> ### Author Response · Authors · 2025-12-03
>
> ### W4: Theorem 3.5 and scale invariance
>
> Thank you for catching this issue. You are right that, there is a typo in our Theorem 3.5.
> This is a **notation/normalization mistake, not a flaw in the method or proof**. In the main text we define DRIC as the *normalized* quantity (8) and show that $Q_\phi$ is invariant under linear rescaling. In Theorem 3.5, however, we inadvertently reused $Q_\phi$ to denote the **unnormalized numerator** and called it “the DRIC value,” which is what causes the contradiction you correctly identified.
>
> In the revised version we
>
> 1. **Introduce explicit notation for the unnormalized variance**
>   $
>    R_\phi := \mathrm{Var}_E\{\mathbb{E}[Y \mid \phi(X),E]-\mathbb{E}(Y\mid E)\},\quad
>    R_\Upsilon := \mathrm{Var}_E\{\mathbb{E}[Y \mid X,E]-\mathbb{E}(Y\mid E)\},
> $
>    and reserve the name “DRIC” only for the normalized ratio $Q_\phi = R_\phi / R_\Upsilon$.
>
> 2. **Correct Theorem 3.5 as follows**
>
>    > **Theorem 3.5 (corrected).**
>    > Suppose $\mathbb{E}[\mathrm{Var}(Y \mid \phi(X))]=0$, so that $Y$ is a deterministic function of $\phi(X)$. Then, under mild regularity conditions,
>    >$
>    > \min_{\phi: \,\mathbb{E}[\mathrm{Var}(Y \mid \phi(X))]=0} R_\phi
>    > \;\ge\; \mathrm{Cov}^2(Y,E).
>    > $
>    > Equivalently, for the **normalized DRIC**
>    >$
>    > Q_\phi = \frac{R_\phi}{R_\Upsilon},
>    > \quad\text{we have}\quad
>    > Q_\phi \;\ge\; \frac{\mathrm{Cov}^2(Y,E)}{R_\Upsilon}.
>    > $
>
>    Here, under rescaling $Y$, both $R_\phi$, $R_\Upsilon$, and $\mathrm{Cov}^2(Y,E)$ is rescaled by the same scale of $Y$; hence the ratio $Q_\phi$ and its lower bound $\mathrm{Cov}^2(Y,E) / R_\Upsilon$ remain **scale-invariant**, consistent with our earlier “Scale-invariant” property.
>
> Importantly, all **experiments already use the normalized DRIC** defined in Eq. (5)/(8), so the empirical results and the qualitative message of Theorem 3.5 are unchanged. This is purely a clarification/typo in notation, and we apologize for the confusion it may have caused.
>
> ###  W5: Weak Simulation
>
> We appreciate this comment. We are sorry for not clearly clarify our merits in the numerical results conclusion. Our goal is to introduce and validate DRIC as a  metric, and invariant learning algorithm.
>
> Importantly, Sec. 4.4 already includes a joint optimization experiment where **DRIC is used as a regularizer**, outperforming IRM, VREx, and ERM in test MSE, thereby confirming its value beyond post-hoc assessment. In addition, Table 17/Fig. 6 show that DRIC correlates monotonically with kernel-based and domain-classifier metrics, offering a normalized and theoretically grounded perspective not available before.
>
> We also want to emphasize that we are the **first to quantify invariance according to the original Definition 2.2**. For example, Table 12 shows a conclusion that is consistent with the IB-IRM paper. Ahuja et al. (NeurIPS 2021 spotlight) argue that their information-bottleneck term “enforces stronger invariance under distribution shift.” **Our empirical DRIC scores are the first to quantify this claim in practice.**  In the revision, we clarify this experimental intent in Sec. 4.

---

### Author Response · Authors · 2025-12-03

## Overview Response to Reviewers and Area Chairs

We thank the reviewers and area chairs for their careful reading and constructive comments.
Below, we (1) briefly restate our main contributions, (2) summarize the key concerns from the reviewers and how we addressed them, and (3) list the concrete manuscript revisions we have made or committed to.

---

## Contributions

Our paper introduces **DRIC**, a **direct, environment-agnostic, and normalized metric** for measuring **representation-level expectation invariance** across environments.

- **Representation-level focus.**
  DRIC targets the stability of  $\mathbb{E}[Y \mid \phi(X), E]$
  across environments, rather than accuracy, marginal dependence, or calibration.

- **Environment-agnostic and normalized.**
  DRIC averages over all environment pairs and is normalized by the identity representation. This makes DRIC values **comparable across datasets and algorithms**, unlike ***raw risk variance, MI/HSIC/MMD, or domain-classifier accuracy***.

- **Theoretical guarantees.**
  We prove (i) **consistency** of the DRIC estimator; (ii) that DRIC = 0 is **equivalent to expectation-level invariance** ; and (iii) an **information-theoretic lower bound** (Theorem 3.5) quantifying the fundamental accuracy–invariance trade-off.

- **Practical utility.**
  We empirically show that DRIC is (i) a robust **post-hoc invariance metric** , and (ii) a **regularizer** that improves OOD performance over ERM/IRM/VREx in joint training.

These contributions are fully aligned with **ICLR’s focus** on **principled, robust representation learning** and out-of-distribution generalization.

---

## Summary of Reviewer Concerns and Our Responses

Several themes appeared across reviewers (ZJ4z, k9K4, jmGM, Rwge); we summarize them jointly here.

### 1. Novelty and relation to existing work

Reviewers asked what precise gap DRIC fills and how it differs from mutual-information / HSIC / MMD penalties and domain-classifier metrics.
In response, we now:

- Explicitly distinguish DRIC from other metrics MI/HSIC/MMD test, domain classifiers and risk variance.
- Provide a **counterexample** where MI/HSIC/MMD and a domain classifier are all zero, yet $\mathbb{E}[Y \mid \phi(X), E]$ still varies with $E$.
- Add a **concise comparison table** contrasting DRIC, MI/HSIC/MMD, domain classifiers, and risk-based criteria.

### 2. Logical consistency around “DRIC = 0”

Reviewers noted an apparent contradiction saying that DRIC = 0 is “unattainable”, and asked whether DRIC = 0 truly implies invariance.

We clarify that:

- At the **population level with overlapping supports**, if $Y \perp E \mid \phi(X)$, then DRIC is **exactly 0** by construction.
- The statement that DRIC = 0 is “unattainable” referred  to the regime of Theorem 3.5 under **perfect prediction**. In that regime, we prove a positive lower bound, so DRIC cannot be 0 unless $Y \perp E$.
  We have now rewritten the text to say explicitly:
  > “DRIC = 0 is unattainable under perfect prediction when $Y$ and $E$ are dependent.”
- When used as a penalty, DRIC is thus a **surrogate objective** that encourages invariance in the same sense that IRM/VREx penalties are surrogates for their ideal conditions.

### 3. Model selection

Reviewers asked how practitioners should trade off DRIC vs accuracy.

We now:

- Connect this to our **robustness experiment** where we vary the proportion of environment-dependent features and observe **monotone changes** in DRIC, matching intuition.
- Add **practical guidance**: monitor effective sample size, clip extremely small density ratios, and treat pervasive near-zero ratios as a diagnostic of severe local support mismatch rather than evidence of perfect invariance.
- Provide **practitioner-facing model selection guidance** (Section 3.3):
 accuracy-constrained selection, target-invariance selection, and scalarization of risk and DRIC, along with reported DRIC ranges observed in our experiments to indicate what “substantially more invariant than ERM” typically looks like.

### 4. Experimental strength

Reviewers asked for more contents on the experiments, noting that DRIC was mainly used as a post-hoc metric.

We clarify that:

- DRIC is already evaluated in three roles:
  1. as a **post-hoc metric** on SEM, CMNIST, and real tabular datasets;
  2. in comparison with **kernel-based invariance tests** and **domain-classifier accuracy**;
  3. as a **regularizer** in joint training (Section 4.4), where DRIC-regularized models achieve the best OOD MSE among ERM/IRM/VREx/DRIC.

---

> ### Author Response · Authors · 2025-12-03
>
> ## Manuscript Revisions (Summary)
>
> In line with the above, we have made or committed to the following revisions:
>
> - Added a **new subsection and comparison table** clearly positioning DRIC relative to other methods.
> - Rewritten the discussion around **DRIC = 0**, Theorem 3.5, and Remark 3.1 to distinguish regimes and make the implications of DRIC = 0 precise.
> - Extended **related work** to cover CLOvE and calibration-based OOD selection, highlighting how DRIC complements these methods.
> - Clarified the **experimental aims** that include metric validation and regularization .
> - Improved **editorial clarity and practitioner guidance**, including standardized references to Definition 2.2 and a short “Practical Guidance” paragraph on interpreting DRIC ranges and trading off DRIC vs accuracy.
> -Added the implementation to extend DRIC to the representation level density-ratio with strong guarantee in Appendix.
> ---
>
> Overall, we believe that the remaining concerns are minor and largely due to initial misunderstandings of what DRIC measures and how it is used. Our clarifications, counterexamples, and strengthened framing resolve these issues and highlight DRIC as a principled, broadly useful tool for invariant representation learning, well within the core scope of ICLR.

---

### Meta-Review · Area_Chair_Dxn8 · 2025-12-31

**Summary:**

Reviews' shared concerns include: 1) lack of more comprehensive evaluation of the proposed metric in high-dimensional benchmark datasets; 2) lack of more careful treatment of the shared support issue (my personal read is that it at least requires further assumptions about the existence of invariant representation and the probabilistic distribution defined on top of it, before defining expectation on top of it); 3) lack of detailed comparison with previous related metrics or methods; 4) lack of analysis on the design choices when the metric is used for model selection and optimization.

In addition, I have two suggestions after reading the paper myself:
1. The evaluation of the domain generalization performance usually include  the average performance for one or more unseen environments, and/or the robust performance, like the performance of worst-case unseen environment. In the current experiment, it seems the test is done on unseen data from seen environments. This should be clarified.

2. There can be potentially a gap between theory and practice, but this should be more rigorously discussed. For example, about the "feasibility" of invariant learning. In practice, domain generalization algorithms are often applied even when there is "disjoint covariate support". Like, many datasets in Domainnet or Domainbed, if we do density (ratio) estimation on raw feature space for different domains, their supports are often disjoint.

**Reviewer Concerns:**

Some of the clarifications are helpful, including more discussion about the comparison with previous metrics. But key questions like broader evaluation are outstanding.

**Reviewer Scores:**

Reviewer jmGM's 6 will probably remain.

Reviewer k9K4's 4 will probably remain. Even though they mention if "partial support handling, stronger metric baselines, or broader DomainBed evaluation, are clarified or strengthened", they are inclined to increase the score, I am not positive the rebuttal resolved all of the issues.

The other reviewers are relatively confident.

So my prediction is this would be a borderline-reject paper after the discussion. So I read the paper and integrate my own opinions.

---

### Decision · Program_Chairs · 2026-01-26

Reject